



# A 4000-year long Late Holocene climate record from Hermes Cave (Peloponnese, Greece)

Tobias Kluge[1,2,3], Tatjana S. Münster[1], Norbert Frank[1], Elisabeth Eiche[3], Regina Mertz-Kraus[4], Denis Scholz[4], Martin Finné[5], Ingmar Unkel[6]

[1]Institute of Environmental Physics, Heidelberg University, 69120 Heidelberg, Germany
[2]Heidelberg Graduate School of Fundamental Physics, Heidelberg University, 69120 Heidelberg, Germany
[3]Institute of Applied Geosciences, Karlsruhe Institute of Technology, 76131 Karlsruhe, Germany
[4]Institute of Geosciences, Johannes Gutenberg University, 55128 Mainz, Germany
[5]Department of Archaeology and Ancient History, Uppsala University, 75126 Uppsala, Sweden
[6]Institute for Ecosystem Research, Kiel University, 24118 Kiel, Germany

*Correspondence to*: Tobias Kluge (tobias.kluge@kit.edu)

**Abstract.**

       The societal and cultural development during the Bronze Age and the subsequent Iron Age was enormous in Greece, however interrupted by two significant transformations around 4200 years b2k (Early Helladic II/III; b2k refers to years before
2000 CE) and 3200 years b2k (end of Late Helladic III). Artefacts and building remains provide some insights into the cultural evolution, but only little is known about environmental and climatic changes on a detailed temporal and spatial scale. Here we present a 4000-year long stalagmite record (GH17-05) from Hermes Cave, Greece, located on Mount Ziria in the close vicinity of the Late Bronze Age citadel of Mycenae and the Classical-Hellenistic polis of Corinth. The cave was used in ancient times, as indicated by ceramic fragments in the entrance area and a pronounced soot layer in the stalagmite.

20         [230]Th-U dating provides age constraints for the growth of the stalagmite (continuous between ~800 and ~5300 years b2k) and the formation of a soot layer (2.5+0.5-0.65 ka b2k). Speleothem $\delta^{18}O$ and $\delta^{13}C$ values together with clumped isotopes and elemental ratios provide a detailed paleoclimate record of the Northern Peloponnese. The proxy data suggest significant centennial scale climate variability (i.e., wet vs. dry). Furthermore, carbonate $\delta^{18}O$ values, calculated drip water $\delta^{18}O$ values, $^{234}U/^{238}U$ activity ratios and elemental ratios suggest a long-term trend towards drier conditions from ca 3.7 to ~2.0 ka b2k.
From 2.0 ka b2k towards growth stop of the stalagmite, a trend towards wetter conditions is observed. A high degree of correlation was found for isotope trends of different speleothems from the Peloponnese and partially with climate records from the Eastern Mediterranean, whereas speleothems and lake records with a larger distance to the Peloponnese show little correlation or even opposing trends.

## 1 Introduction

30         Southern Greece saw significant societal and cultural changes during the Bronze Age and Early Iron Age, which seem to have happened "rapidly" on a scale of centuries or even decades (Bintliff, 2012; Drake, 2012; Finné et al., 2017). This is





documented by variation in agricultural, mining, and forging techniques, pottery and clothing style, government structure, religious practise, or trade details (e.g., Drake, 2012; Finné et al., 2017; Middleton, 2012). In contrast, there is currently only limited knowledge of environmental and climatic changes in the region that are often determined using geological archives (e.g., Finné and Weiberg, 2018). Whereas single events may be known by the day in archaeological findings, temporal

resolution is typically on annual, decadal or even longer scales for geological archives. Furthermore, paleoclimate archives such as lake sediments or speleothems should be located in close proximity to archaeological sites to allow a meaningful comparison of climatic and cultural changes. However, in Southern Greece this is rarely the case due to the scarcity of suitable paleoclimate records compared to the abundance of archaeological sites (Weiberg et al., 2016). Since the review article of Weiberg et al. (2016), several paleoclimate studies from Greece with varying resolution and temporal coverage have been

published and provide records from caves (Finné et al., 2017), lagoons (Katrantsiotis et al., 2018, 2019) and lakes (Seguin et al., 2019). A paleoclimate record from the region with sufficiently high resolution (annual to decadal) completely covering the Aegean Bronze Age and Iron Age is so far not available.

Here we focus on speleothems as paleoclimate archives for the Peloponnese and compare our record from Hermes Cave to other regional archives, notably speleothems and lake sediments from the Peloponnese, including a sediment core

from Lake Stymphalia located 10 km south of the cave site (Heymann et al., 2013; Seguin et al., 2019). Speleothems provide the possibility of precise dating (up to permil precision, i.e., ±10 years at an age of 10,000 years; Cheng et al., 2013) and a wealth of proxy information (e.g., Fairchild and Baker, 2012). In addition to traditional proxies (elemental ratios, oxygen and carbon isotope ratios), we also determined carbonate clumped isotope values ($\Delta_{47}$; Eiler, 2007) at key periods for quantification of the proxy information and for disentangling the different environmental/climatic parameters in the multi-proxy space.

Carbonate clumped isotopes refer to carbonate molecules that contain both $^{13}C$ and $^{18}O$ (Eiler, 2007). Their abundance relative to a pure stochastic distribution is almost completely governed by temperature in the case of equilibrium mineral formation and is mass-spectrometrically quantified as $\Delta_{47}$ (Wang et al., 2004). The $\Delta_{47}$ value increases with decreasing temperature and has a temperature sensitivity of about 0.003 ‰/°C at Earth surface conditions. In stalagmites, $\Delta_{47}$ measurements can also be used to determine potential contributions of kinetic isotope fractionation to the proxy signals (Kluge

and Affek, 2012; Kluge et al., 2013) and, using independently derived climate information, to correct back to the equilibrium conditions (e.g., Wainer et al., 2011; Kluge et al., 2013).

The proximity of various paleoclimate records on the Peloponnese (albeit with different resolution and temporal gaps) additionally allows for a cross check of the proxy interpretation on local and regional scale. For example, the new data from Hermes Cave helps to assess if the whole peninsula was continuously affected by the same climate systems or if a significant

and persistent divide existed for certain time periods (as, for instance, suggested by Katrantsiotis et al., 2019). Our study revealed that there occurred a long-term trend of decreasing rainfall from ca. 4.0 to 2.0 ka. This trend was preceded at ca. 4.2-4.0 ka by a pronounced high-amplitude fluctuation between a wet and a dry state, that are related to the most and the least negative δ$^{18}$O values of the whole record, respectively.



## 2. Study Area

Hermes Cave is located in the northern Peloponnese at the eastern part of the central mountain range, about 10 km north of Lake Stymphalia and ca. 50 km from Corinth, Mycene, or Argos (Fig. 1). According to mythology, it is the birthplace of the Greek God Hermes, son of Zeus and Maia, one of the Pleiades. Archaeological artefacts such as pottery and small figurines indicate that the cave was used as a place of cultural devotion since the 8th century BCE. However, most of the ancient visitors seem not to have penetrated more than 50 m into the cave to a depth of max. 30 m (Kusch, 2000). The cave entrance at the Eastern slope of Mount Ziria (Kyllini) is situated at 1614 m above sea level. It extends for about 210 m into the mountain following the sedimentary layering and tectonic structure of the host rock (Fig. 1b) to a depth of about 72 m below the entrance level (Fig.2) (Kusch, 2000). The entrance of the cave is located on a steep slope facing into a deep valley covered with coniferous vegetation. Today, the soil cover in the area above the cave is thin and patchy, revealing in many places the barren karstified Upper Triassic to Lower Cretaceous limestone belonging to the Gavrovo-Tripoli Zone (Fig. 1b, Nanou and Zagana, 2018). Vegetation mainly consists of spruces, shrubs and herbaceous plants. Temperature was measured during retrieval of the stalagmite (GH17-05) and was 9.2 °C in the deepest part of the cave and 9.0 °C close to the former position of the stalagmite (at about 55 m depth). The relative humidity of the cave air was >92 % during the sampling visit. $CO_2$ of cave air was measured to 4300 ppmV in the deepest part and to 3270 ppmV close to the collected stalagmite. The drip site feeding the stalagmite was active at the time of the collection. The surface of the stalagmite was wet and covered with white calcite crystals possibly indicating recent calcite growth precipitation.

Annual precipitation at Mount Ziria amounts to ~1000-1300 mm (Voudouris et al., 2007; Nanou and Zagana, 2018) and is strongly different from the much lower annual precipitation to the east (e.g., 400-600 mm in Athens; IAEA-GNIP). Based on daily precipitation data recorded between 1949 and 2011 at the meteorological station Driza (Greek Special Secretariat for Water, Ministry of Environment and Energy), Seguin et al. (2019) calculated a mean annual precipitation of 618±201 mm at Lake Stymphalia, with a high inter-annual variability during this period. The region receives most precipitation during winter time (October – March) with no or very little effective infiltration during summer time (Fig.3 a, b). The IAEA GNIP stations located in Athens show a slightly negative correlation between rainfall $\delta^{18}O$ values and rainfall amount (Supplementary Fig. S1), which is consistent with observations in the Eastern Mediterranean (Fig. 3c). Other effects such as moderate seasonal shifts in infiltration (up to 50 % in winter and summer season, respectively) cause minor changes in the annual average rainfall $\delta^{18}O$ value (Supplementary Fig. S2a). Assessing infiltration changes by moderately varying the mean annual temperature (±3°C) leads to negligible changes in mean annual infiltration water $\delta^{18}O$ (Supplementary Fig. S3). Uniform infiltration increases throughout the year have a larger potential for modifying the mean $\delta^{18}O$ values of annual infiltration, but still only yield changes of ca. 0.1 ‰ for 50 % relative increase in annual rainfall and slightly higher effects for corresponding reduction (Supplementary Fig. S2b). Infiltration during snow melt is very efficient and has an over-proportional contribution relative to the total precipitation (Earman et al., 2006). As alteration of the $\delta^{18}O$ values of the snow on the surface happens (exchange with atmospheric vapour), residual snow approaches much higher $\delta^{18}O$ values compared to the fresh snow





and therefore masks its effect on the average recharge $\delta^{18}O$ value (Earman et al., 2006). An assessment of the snowmelt contribution on the averaged infiltration water $\delta^{18}O$ is therefore difficult. The overall strongest effect on the $\delta^{18}O$ value of the infiltration is likely the amount effect that causes a negative shift of 1 ‰ per 200-300 mm annual rainfall increase (Fig. 3c; Bar-Matthews et al., 2003). Given this relationship is also valid for Mount Ziria it could be used to transfer average rainfall

$\delta^{18}O$ values into changes of the rainfall amount.

## 3. Materials and Methods

Stalagmite GH 17-05 from Hermes Cave is about 6 cm long with a layered structure consisting of transparent and whitish laminae on the mm and sub-mm scale (Fig. 4). A significant shift in appearance is visible at ca. 45 mm from top showing no visible layering and an additional change in colour (brownish appearance) at ca. 50 mm from top. Between 45 and

50 mm from top, the stalagmite shows increased porosity. A soot layer is found in the upper part of the stalagmite at 15 mm from top. The single occurrence of the soot layer in stalagmite GH17-05 asks for understanding its connection to environmental/climatic changes or variations in the number of visitors linked to the Hermes cult.

### 3.1 Dating:

For $^{230}Th/U$ dating, ten thin rectangular samples were taken along visible growth layers perpendicular to the growth

axis using a diamond band saw (Table 1). Each of the samples had a thickness of about 2 mm and a weight of 120-240 mg. The sample processing followed the protocol developed by Wefing et al. (2017) and was adopted to speleothems as described in Warken et al. (2018). In brief, the samples were manually pre-treated to obtain pure carbonate material, dissolved in acid and spiked with artificial Th and U isotopes ($^{229}Th$, $^{233}U$, $^{236}U$). Subsequently, the solution was passed through an ion exchange column (UTEVA resin) to purify U and Th. The measurements were done using a multi-collector inductively coupled plasma

source mass spectrometer (Thermo Scientific Neptune$^{Plus}$) at Heidelberg University equipped with a desolvator (CETAC Aridus) and an auto-sampler (Elemental Scientific SC-2 DX). Measurement protocols and subsequent correction of the measured activity ratios followed Warken et al. (2018). The absolute accuracy was determined with the standard-sample bracketing technique using the Harwell Uraninite HU-1 secular equilibrium standard. The corrected isotope ratios were then used to calculate U-series ages according to the decay equations. The error propagation accounts for the statistical uncertainties

and for detrital $^{232}Th$-correction. Ages have been corrected for a residual non-carbonate (detrital) contamination of $^{230}Th$ with the $^{232}Th$ concentration using a ($^{232}Th/^{238}U$) activity ratio of 0.521 (i.e., a ($^{230}Th/^{232}Th$) activity ratio of 1.92±0.96) and assuming secular equilibrium of the detritus. The correction factor for detrital correction was determined using the procedure of Budsky et al. (2019a) and is based on varying the ($^{232}Th/^{238}U$) activity ratio of the detritus in order to minimize the number of age inversions observed in the chronology of the corrected age data. U-series results and ages are reported relative to the year 2000

and labelled as b2k (before 2000 CE). If not indicated otherwise, "ka" refers to "ka before 2000 CE (b2k)" throughout the manuscript.

### 3.2 Stable isotopes





Carbonate powder for stable isotope analysis ($\delta^{13}$C, $\delta^{18}$O) was retrieved using a micro-mill, where the tracks followed the growth layers. Powders were sampled at 166 μm steps, reacted with phosphoric acid (103 %) at 72 °C in a gas-bench system and measured on a Delta V advantage mass spectrometer at the Karlsruhe Institute of Technology (KIT). All calcite $\delta^{18}$O and $\delta^{13}$C values are reported relative to VPDB and water relative to VSMOW. Each sample measurement consists of ten repetitions leading to a standard deviation of $\leq 0.08$ ‰ for $\delta^{13}C_{VPDB}$ and $\leq 0.12$ ‰ for $\delta^{18}O_{VPDB}$. The quality of the measurements was checked by regularly including Carrara marble (in-house reference material) into the measurement procedure (n=119). The achieved accuracy was ±0.03 ‰ ($\delta^{13}C_{VPDB}$) and ±0.08 ‰ ($\delta^{18}O_{VPDB}$), respectively.

### 3.3 Elemental analysis

Analyses were performed in line-scan mode at the Institute of Geosciences, JGU, Mainz, Germany, using an ESI NWR193 ArF excimer laser ablation system equipped with the TwoVol$^2$ ablation cell, operating at 193 nm wave length, coupled to an Agilent 7500ce quadrupole ICP-MS. Prior to each line scan, surfaces were pre-ablated to prevent potential surface contamination. For analyses, line scans were carried out at a scan speed of 10 μm/s using a rectangular beam of 130 μm × 50 μm (beam for pre-ablation was 50 μm × 100 μm). Laser repetition rate was 10 Hz, and laser energy on the samples was about 3.4 J/cm$^2$. Background intensities were measured for 15 s. Monitored isotopes included $^{25}$Mg, $^{26}$Mg, $^{27}$Al, $^{31}$P, $^{43}$Ca, $^{55}$Mn, $^{56}$Fe, $^{57}$Fe, $^{86}$Sr, $^{88}$Sr, $^{135}$Ba, $^{137}$Ba, $^{138}$Ba, and $^{208}$Pb. The calcium carbonate reference material USGS MACS-3 was used to calibrate element concentrations applying values available from the GeoReM database (http://georem.mpch-mainz.gwdg.de/, compare also Jochum et al., 2005, 2011, 2012). Quality control materials (QCMs) (USGS BCR-2G, NIST SRM 610 and 612) were used to monitor the LA-ICP-MS analysis and calibration strategy. QCMs were assessed by measuring 300 μm long line scans corresponding to 30 s acquisition time. Element concentrations determined for the QCMs had a precision of <5 % (NIST 612 and MACS3) and <8% (NIST 610 and BCR-2G) (1 SD). Signals of all measurements were monitored in time-resolved mode and processed using an in-house Excel spreadsheet. Calculations included an initial baseline subtraction, normalisation of all isotopes to $^{43}$Ca, and removal of outliers (details are given in Mischel et al., 2017a).

### 3.4 Clumped isotopes

Sample preparation and analysis followed the methods as described in Kluge et al. (2015) and Weise and Kluge (2019). CaCO$_3$ powder was extracted using a dental drill at 8 key segments at the growth axis. Samples of 3.0-3.5 mg were reacted in individual reaction containers with 105% phosphoric acid (~ 1 ml per sample) for 10 min at 90°C under continuous stirring. The reaction containers were carefully evacuated prior to reaction ($10^{-1}$ to $10^{-2}$ mbar). The emerging CO$_2$ was continuously collected with a liquid-N$_2$ cooled trap and cleaned using a procedure initially described by Dennis and Schrag (2010). Volatile gases were cryo-distilled at liquid nitrogen temperature. Subsequently, water was separated from the remaining gas using a dry-ice ethanol cooled trap. The water-free CO$_2$ gas was then passively passed through silver wool and another trap densely packed with Porapak Q held at -35 °C. The cleaned CO$_2$ gas was directly transferred to the mass spectrometer and analyzed immediately or within a few hours.

Isotopic measurements were performed at the Institute of Environmental Physics using a multi-collector isotope ratio mass spectrometer (Thermo Scientific MAT 253Plus) with a baseline monitoring cup on m/z 47.5 and $10^{13}$ Ω resistors on m/z





47-49. The analysis protocol followed the procedures described by Huntington et al. (2009) and Dennis et al. (2011) (8 acquisitions with 10 cycles each, integration time for each cycle: 26 s). Each acquisition included a peak center, background measurements and an automatic bellows pressure adjustment aiming at a 6 V signal at mass 44. The first acquisition additionally included a recording of the sample m/z 18 (water vapour residual) and m/z 40 signal (Ar – indicator for air

residual). The m/z 47-49 signals are influenced by a negative background potentially induced by secondary electrons and broadening of the m/z 44 peak (He et al., 2012; Bernasconi et al., 2013; Fiebig et al., 2015). For each cycle, the baseline signal on m/z 47.5 was therefore measured simultaneously to the actual sample and reference gas analysis on m/z 44-49. The 47.5 cup only records background and is therefore sensitive to secular changes of the baseline on the short (seconds) as well as longer term (hours to weeks). For pressure-baseline correction (PBL), high-voltage peak scans were manually taken at the

beginning and/or end of a measurement run (integration time 0.5 s, step size 0.0005 kV). The background was determined via high-voltage scans and adjusting the m/z 44 signal by increasing or decreasing the bellows pressure. The working pressure for the measurement run was typically about 22 mbar for a m/z 44 signal of 6000 mV.

All data were evaluated with an in-house program that includes a PBL correction based on the m/z 47.5 signal. The 47.5 baseline signal was correlated to the baseline of all other cups and therefore allows a correction of each individual data

point. The empirical transfer function (ETF) was determined based on the m/z-47.5-corrected $\Delta_{47}$ values and uses carbonate reference materials (In house marble "Richter", NBS 19, ETH1-ETH4; Meckler et al., 2014; Müller et al., 2017)  and water-equilibrated gases (5 °C, 25 °C, 90 °C) with agreed $\Delta_{47}$ values as reference (Dennis et al., 2011). The sample gas was measured against an in-house reference gas ($\delta^{13}C$ = - 4.42 ‰ VPDB, $\delta^{18}O$ = -9.79 ‰ VPDB). Updated isotope parameters following Brand et al. (2010) were used for the $\Delta_{47}$ calculation (Daëron et al., 2016; Schauer et al., 2016).

## 4. Results

### 4.1 Th/U dating

All carbonate samples yielded relatively low $^{238}U$ concentrations (220-160 ng/g) and detrital $^{232}Th$ in general below 4 ng/g (Table 1). The ($^{230}Th/^{232}Th$) activity ratios range from 70.6 to 2.0 (Table 1). According to Richards and Dorale (2003), a correction for detrital contamination may be necessary if the measured ($^{230}Th/^{232}Th$) activity ratio is <200. This suggests a

substantial degree of contamination for several samples of stalagmite GH 17-05. The lowest value and, thus, the highest degree of contamination is observed for the topmost sample. In order to constrain the ($^{230}Th/^{232}Th$) activity ratio of the detritus, we applied the algorithm of Budsky et al. (2019a), which varies the ($^{232}Th/^{238}U$) activity ratio of the detritus to minimise the number and sum of age inversions observed in the corrected age data (Fig. S4). This procedure resulted in a ($^{230}Th/^{232}Th$) activity ratio of 1.92 ± 0.96 for the contaminating phase. For this value, no inversions are observed, strongly suggesting that it

is appropriate to account for detrital contamination of the Hermes Cave stalagmite. We note that the determined ($^{230}Th/^{232}Th$) activity ratio of the detritus is substantially larger than the commonly used bulk Earth value of 0.8. However, values in this range and even higher have been observed for various other speleothems (e.g.; Beck et al., 2001; Hellstrom, 2006; Fensterer





et al., 2010; Rivera-Collazo et al., 2015; Budsky et al., 2019a, b; Warken et al., 2019) are, thus, not uncommon. The correction (i.e., the difference between the corrected and the uncorrected age) ranges from 0.1 to 4.1 ka (Table 1). We assume an uncertainty of $\pm 50$ % for the detrital ($^{230}$Th/$^{232}$Th) activity ratio, which is propagated to the corrected ages. This results in relatively large uncertainties for some the corrected ages, which were used to construct the age model. The effect of the

correction is, thus, accounted for by the uncertainties of the age model.

Extracted samples date between $4.29 \pm 0.62$ ka and $0.2 \pm 1.2$ ka and with typical uncertainties of the corrected ages generally between 0.07 and 0.29 ka; higher in the case of significant detrital correction from $\pm 0.5$ to $\pm 1.2$ ka (Fig. 4). A chronology was established with a Bayesian age-depth-modelling using the R package Rbacon (v.2.3; Blaauw and Christen, 2011) and cross-checked with StalAge (Scholz and Hoffman, 2012). Both approaches yield generally consistent chronologies

(Suppl. Fig. S5). In the following, we refer to the Rbacon chronology. Linear extrapolation of the chronology with an average growth rate suggests an age of $0.8_{+1.0-1.4}$ ka for the top of the stalagmite, within uncertainty consistent with the active drip site and the possibility of recent calcite formation. Note that there is visual indication of a thin top layer distinct from the older parts below. We hesitate to extrapolate the chronology towards the bottom, as there is a clear change in appearance at 49 mm from top with a colour change from whitish towards brownish layers (Fig. 4). At 49 mm from top, the Rbacon model suggests

an age of the stalagmite of $5.3_{+1.0-0.7}$ ka.

### 4.2 Stable isotopes

The calcite $\delta^{18}$O values vary between -6.2 and -7.4 ‰ with the least negative values being found around 2.0 ka and 4.2-4.1 ka (Fig. 5, data in supplementary file). Overall, there is a long-term trend from the most negative $\delta^{18}$O values prior to 4.0 ka (oldest part evaluated for oxygen and carbon isotopes) towards the least negative values at 2.0 ka, interrupted by a rapid

high amplitude fluctuation at around 4.2-4.0 ka. The youngest part of the stalagmite shows a clear trend towards more negative $\delta^{18}$O values until growth cessation.

The carbon isotope ratios exhibit no pronounced long-term trend, only a tendency towards less negative values in the topmost part of the stalagmite (last 400 years of growth). In general, the $\delta^{13}$C values are characterized by high amplitude short-term fluctuations between -8 and -10 ‰. The least negative $\delta^{13}$C values are observed in the top part.

### 4.3 Elemental ratios

Al, Mn, and Fe are episodically above background and show largely correlated signals (Fig. 6, periods younger than 1.5 ka), indicating a common source for particulate input. P/Ca ratios (Fig. 5) fluctuate on a ca. bi-centennial scale (20 peaks between 4.3 and 0.8 ka) and also shows a long-term trend towards higher ratios. A similar number of peaks within the same time period are also found for the $\delta^{13}$C (18) and $\delta^{18}$O values (20). Ba/Ca and Sr/Ca ratios are relatively constant with slightly

elevated values at periods with elevated Pb/Ca and Mn/Ca ratios (Fig. 6). Mg/Ca is largely uncorrelated to the other elemental ratios (Supplementary Fig. S6) and shows significant variations on a less regular scale (Fig. 5, 6). Note that elements and C and O isotopes were not measured on exactly the same track.

### 4.4 Clumped isotopes and calculated water $\delta^{18}$O values



Clumped isotope $\Delta_{47}$ values range from 0.728±0.030 ‰ to 0.775±0.020‰, corresponding to temperatures of 2 to 14 °C (Table 2). The mean value of the eight isotope samples (with 2-3 replicates per sample) is 0.749 ± 0.014 ‰, corresponding to a temperature of 8.5 ± 3.9 °C, well overlapping with the current cave temperature of 9.0 °C at the former location of the stalagmite GH17-05. $\Delta_{47}$ values and related temperatures agree all (with one exception at 37 mm depth, ~4.4 ka) within uncertainty with the current cave temperature. Due to the measurement uncertainties of 1-6 °C, trends in temperature cannot be inferred. However, the general correspondence of individual and the average $\Delta_{47}$-based temperature with modern measurements suggest no or negligible kinetic isotope effects that would cause overestimated temperatures.

Using the $\Delta_{47}$-based temperatures we calculated $\delta^{18}O$ values of dripwater using the fractionation factor $^{18}\alpha(H_2O$-calcite) of Kim and O'Neil (1997) and the corresponding calcite $\delta^{18}O$ values. The calculated $\delta^{18}O$ values of the dripwater are between -7.3 and -10.4 ‰ and follow the trend of the calcite $\delta^{18}O$ values with the least negative values around 4.1 ka and 3 ka and the most negative values around 4.4 ka (Supplementary Fig. S7).

## 5. Discussion

### 5.1 Paleoclimatic interpretation of the GH17-05 proxy data

In the paleoclimatic interpretation we particularly focus on the time period with the strongest chronology (e.g., around the 4.2 ka event). Note that age uncertainties at the stalagmite top are elevated, making there a direct comparison with historical events challenging. The uncertainty of the stalagmite chronology should also be taken into account when discussing archaeological findings with the paleoclimatic record.

*Negligible disequilibrium isotope fractionation and prior calcite precipitation*

For meaningful interpretation of the speleothem proxy data, knowledge of potential disequilibrium isotope effects or kinetic isotope fractionation (e.g., Mickler et al., 2004; Kluge and Affek, 2012; Affek et al., 2014) is essential. Disequilibrium effects can be related to Prior Calcite Precipitation (PCP), i.e. when the percolating, supersaturated karst water causes carbonate precipitation before reaching the stalagmite (e.g., Fairchild and Treble, 2009; Borsato et al., 2016). The chemical and isotopic evolution of a thin solution film on the top and the flanks of a stalagmite can also cause disequilibrium (e.g., Scholz et al., 2009; Dreybrodt and Scholz, 2010; Hansen et al., 2019). A particularly sensitive indicator for disequilibrium effects is the clumped isotope $\Delta_{47}$ value (Kluge and Affek, 2012), in addition to the commonly used Hendy test (Hendy, 1971). The $\Delta_{47}$ value would deviate significantly towards lower values (i.e. towards higher apparent temperatures) if disequilibrium conditions prevailed. The general agreement of the calculated temperatures (based on the $\Delta_{47}$ values measured in GH17-05, Table 2) with the current cave temperature suggests no or very limited influence of disequilibrium effects or PCP. Limited or non-existing PCP is also consistent with the findings of Borsato et al. (2016), who suggested PCP to be relevant below an elevation of 1200 m in a similar environment due to more frequent periods of non-infiltration and opportunities for partially air-filled epikarst space. Hermes Cave is situated in the high montane to subalpine zone at a higher elevation of around 1600 m. Further evidence



for the absence of PCP comes from missing or insignificant correlations between Mg/Ca, Sr/Ca and $\delta^{13}$C. Thus, given the non-measurable kinetic effect and the likely absence of PCP, the calcite $\delta^{18}$O values should directly reflect the dripwater $\delta^{18}$O values with the corresponding temperature dependent fractionation $^{18}\alpha$(H$_2$O-calcite). Calculated dripwater $\delta^{18}$O values (Table 2, section 4.4) vary around the estimated rainfall $\delta^{18}$O values of -7.5 to -9.3 ‰ for the Killini/Ziria mountain range (Bowen, 5  2019; isomap.edu), corroborating that disequilibrium effects and PCP likely play no or an insignificant role for the proxy interpretation.

*Oxygen isotope ratios*

Transferring the calcite $\delta^{18}$O values into dripwater $\delta^{18}$O values (by assuming an approximately constant cave temperature and no or little kinetic isotope fractionation due to degassing at the drip point) we can derive relative changes that can be linked to past variations infiltration and rainfall variations. A long-term trend towards less negative calcite and related 10  dripwater $\delta^{18}$O values is observed from 4.0 ka to ca. 2.0 ka, followed by a slightly more rapid decrease to more negative $\delta^{18}$O values in the youngest part of the stalagmite (Fig. 5, Table 2). Only considering the long-term signal and its trend (disregarding higher frequency fluctuations), maximum and minimum values deviate by about 0.4 ‰ during this time period. This could reflect either a small shift in temperature of about 2 °C, thereby modulating $^{18}\alpha$(H$_2$O-calcite) by about 0.4 ‰, or, if temperature remained constant, a change in the amount of rainfall and infiltration (about 80-100 mm/year based on an eastern 15  Mediterranean relationship; e.g. Bar-Matthews et al., 2003). The long-term trend is overlain by higher-frequency fluctuations with about 20 peaks that yield amplitudes > 0.2 ‰ in the interval from 4.3 ka to the stalagmite top (average periodicity ~180 years). Outstanding is one high-amplitude change at 4.2-4.0 ka (Fig. 7) that shows the largest change of the whole record with a 1.2 ‰ shift within about 60-70 years and includes both the least and the most negative calcite $\delta^{18}$O values (discussed in more 20  detail in section 5.2). The cave temperature should not have varied substantially within this rather short time period (due to the slow thermal diffusivities and heat capacities of the karst rocks). Therefore, the signal can mainly be attributed to changes in the hydrological cycle transferred to the stalagmite via rainfall and infiltration. For the same reason, the other observed high-frequency variations beyond 0.2 ‰ may also be related to significant changes in infiltration amounts with dry phases in case of less negative $\delta^{18}$O values and wet conditions at time periods with negative $\delta^{18}$O values.

*Carbon isotopes and P/Ca ratios*

25  The interpretation of the calcite and calculated dripwater $\delta^{18}$O values (Table 2) is corroborated by the $\delta^{13}$C values and the P/Ca ratios (Fig. 5). The $\delta^{13}$C values show no significant long-term trend, but high-frequency fluctuations with about 18 peaks with an amplitude beyond 0.5‰ from 4.3 ka to the stalagmite top (average periodicity ~190 years). $\delta^{18}$O and $\delta^{13}$C values are weakly anti-correlated, i.e. more negative $\delta^{13}$C values correspond to less negative $\delta^{18}$O values (Supplementary Fig. S8). 30  This anti-correlation is best visible for a few case examples, e.g., around 4.2-4.0 ka. The most negative $\delta^{13}$C values occur together with the least negative $\delta^{18}$O values and the corresponding positive peak in the $\delta^{13}$C values (about 2 ‰ above the minimum) matches directly the most negative $\delta^{18}$O value of the record. Calcite $\delta^{13}$C values can be influenced by various





factors, whereof we exclude PCP for the Hermes Cave stalagmite due to the missing kinetic signal in $\Delta_{47}$ and no correlation between Sr/Ca and Mg/Ca (Fig. S6). The observed anti-correlation between the $\delta^{18}O$ and $\delta^{13}C$ values is relatively unusual for speleothem calcite, but has also been reported in speleothems from Soreq Cave (Israel) (Bar-Matthews et al., 1999, 2003). There, a significant anti-correlation between $\delta^{18}O$ and $\delta^{13}C$ values is recorded during Sapropel events S1 and S5, with the least

negative $\delta^{13}C$ values occurring during the wettest period with most negative $\delta^{18}O$ values. As possible explanation, Bar-Matthews et al. (2003) suggested the stripping of the soil cover by deluge events resulting in water reaching the stalagmites after only little interaction with soil $CO_2$. This could be even more important at the high-elevation Hermes Cave site with relatively thin and patchy soil cover. The reduced interaction of infiltrating water with the soil zone during wet periods due to surface runoff and preferential localized infiltration may also be the likely reason for the weak $\delta^{18}O$-$\delta^{13}C$ anti-correlation found

in the Hermes Cave stalagmite. The variation of the P/Ca ratio corresponds to that of the $\delta^{13}C$ values with higher P/Ca ratios generally matching less negative $\delta^{13}C$ values (Fig. 5) with the exception from ca. 3.9 to 3.4 ka. Increased P/Ca ratios during wet periods (see also Mischel et al., 2017b) (coinciding with more negative $\delta^{18}O$ and less negative $\delta^{13}C$ values in GH17-05) are potentially due to particle erosion from the soil cover (e.g., Kronvang et al., 1997, 1999) and to a minor degree due to leaching. Fe/Ca ratios support this hypothesis at 4.1-4.0 ka with a peak coinciding with higher P/Ca ratios and more negative

$\delta^{18}O$. Similarly, Fe/Ca (and to some degree Al/Ca) ratios are elevated simultaneously with more negative $\delta^{18}O$ values from 1.5 to 0.8 ka (Fig. 6). In contrast, during dry periods large fractions of the available phosphorous is taken up by plants and therefore causes reduced P/Ca ratios. The corresponding correlation of higher phosphorus concentration at elevated rainfall was also found by Treble et al. (2003) based on a high-resolution analysis of a recent stalagmite. In summary, we associate calcite with more negative $\delta^{18}O$ values with wet periods. $\delta^{13}C$ values and elemental ratios are likely influenced by the associated elevated

rainfall that reduces interaction with the soil zone (by fast preferential infiltration through sinkholes, fractures, etc.) and causes soil erosion including particulate transport of phosphorous and other elements.

*Elemental ratios*

Regarding elemental ratios, mostly Mg/Ca and to a minor degree Sr/Ca, P/Ca, and Ba/Ca or other elemental ratios have been used for extracting paleoclimate information from speleothems (e.g., Huang et al., 2001; Treble et al., 2003;

Fairchild and Treble, 2009). Mg/Ca is the most widely used elemental ratio thought to generally reflect paleo-hydrological changes (Fairchild and Treble, 2009; Warken et al., 2018). The Mg/Ca ratio can be modified by PCP (Sinclair et al., 2012) or changed through dilution under high karst-water flow and by source changes from matrix seepage to more direct shaft flow. Sr/Ca and Ba/Ca often co-vary and were found to be strongly influenced by speleothem growth rate (Treble et al., 2003) that could also be used in some cases as an indicator for annual lamination (Warken et al., 2018). Ba/Ca and Sr/Ca are correlated

with each other in GH 17-05, but uncorrelated to Mg/Ca (Fig. 6, Supplementary Fig. S6). The missing correlation between Mg/Ca and Sr/Ca, in addition to the observation that clumped isotopes reflect cave the temperature, suggests that PCP can largely be excluded as a driver for proxy variability in GH17-05. The only correspondence between Ba/Ca and Mg/Ca is found at about 4.2-4.0 ka with a peak towards elevated Mg/Ca and Ba/Ca ratios, pointing to an extraordinarily strong forcing (see





section 5.2). With the exception of another peak at ca. 1.3 ka in Ba/Ca and at ca. 1.5 ka and 0.8 ka in Sr/Ca ratios, both ratios are relatively constant on the long-term (Fig. 4). Mg/Ca ratios show a long-term trend similar to that of the calcite $\delta^{18}$O values with lowest ratios at 2.0-1.5 ka. This long-term trend with a relative change in ratios of about 15-20 % may be related to a temperature change of 2-5 °C (using the temperature dependence of the partition function of Huang and Fairchild, 2001).

However, the temperature-dependence of the partition function seems to be subordinate relative to other effects, notably hydrological factors (Fairchild and Treble, 2009). One of the highest Mg/Ca ratios is found at 4.0 ka coinciding with the most negative $\delta^{18}$O values. The correspondence of high Mg/Ca ratios with more negative $\delta^{18}$O suggests a major hydrological influence on both values, i.e., wet conditions at those time periods. High Mg/Ca ratios at periods with negative $\delta^{18}$O values and increased infiltration is uncommon for speleothem records as high Mg/Ca ratios are normally indicative for an extended

contact time with the aquifer rock during dry periods (Fairchild et al., 2000). As visible in $\delta^{13}$C values and P/Ca ratios, heavy rainfall events and related erosion with elevated soil particle flux could explain this unusual negative correlation between Mg/Ca and $\delta^{18}$O values. In the related time period around 4.0 ka the particle-sensitive ratio Fe/Ca and at 1.5-0.8 ka Fe/Ca, together with the particle-sensitive ratios Mn/Ca and Al/Ca are elevated (Fig. 6).

Additional supporting evidence for the long-term trend in rainfall and infiltration with generally wetter conditions

prior to 4.2 ka and towards the stalagmite top and drier conditions between ca. 3.5 and ca. 2.0 ka comes from $^{234}$U/$^{238}$U activity ratios (Supplementary Fig. S7). Higher $^{234}$U/$^{238}$U activity ratios are consistent with the least negative $\delta^{18}$O values of GH 17-05 in the same time period. High activity ratios are observed prior to 4.2 ka and towards the stalagmite top. The U activity ratios decrease from about 4.3 ka to ~2.8 ka, where they reach a minimum. Following Frumkin and Stein (2004), higher activity ratios are indicative of selective $^{234}$U removal from the soil, supporting our interpretation of increased wetness and potentially

heavy rain events during these periods.

In summary, we use the evolution of the calcite $\delta^{18}$O values as primary indicator for wet and dry periods with the most negative values representing wet periods. We suggest that $\delta^{13}$C values and trace elements are strongly influenced by intense rainfall events causing reduced water interaction with soil $CO_2$ (i.e., more positive $\delta^{13}$C during deluge periods) and soil erosion with transport of particulate matter (increased Mg/Ca and $^{234}$U/$^{238}$U ratios during wet periods with peaks in Al/Ca,

Mn/Ca and Fe/Ca).

### 5.2 Observations at 4.2-4.0 ka

Several studies in the Middle East and the Mediterranean region suggest significant climatic changes around 4200 cal BP = 4250 b2k (see e.g., Rousseau et al., 2019). In particular, based on arboreal pollen records, a significant forest decline is visible in the central Mediterranean at 36°-39°N and for many sites at 39-41°N (Di Rita and Magri, 2019). In the Levant and

the Central Mediterranean the climatic conditions seem to be drier around 4.5-4.1 ka BP compared to earlier or later periods



(Kaniewski et al., 2018; Isola et al., 2019). It is hypothesized that a northward shift of the North-African high-pressure system caused the observed changes (Di Rita and Magri, 2019). Simultaneously, an intensification of precipitation is observed in the Southern Alps (Cartier et al., 2019), potentially due to an atmospheric blocking regime related to a weakened subpolar gyre (Jalali et al., 2019). At the same time, a strengthening of the Siberian High is suggested to result in reduced precipitation in

South-Eastern Europe (Perşoiu et al., 2019). A conclusive picture over the complete Mediterranean region, however, has not emerged yet, suggesting that patterns may be regional or that the resolution of many records is not sufficient to resolve the related oscillations (Bini et al., 2019; Finné et al., 2019).

In our Hermes Cave record, the highest fluctuations in the $\delta^{18}$O values are found between 4.15 and 4.02 ka ($\pm$0.2 and $\pm$0.3 ka, respectively) (corresponding to 4100-3970 cal BP in $^{14}$C based chronologies), which is consistent with the timing of

an aridity event in Northern Mesopotamia within the given uncertainty ranges (Carolin et al., 2019). The amplitude of this fluctuation in GH17-05 exceeds 1 ‰ and includes both the most negative and the least negative $\delta^{18}$O value of the entire record (Fig. 7). Notably, this rapid and significant variation is followed by a second fluctuation from 4.0 to 3.85 ka (amplitude: 0.7 ‰). Significant changes during the same time period are also visible in the $\delta^{13}$C values, Mg/Ca and P/Ca (Fig. 5), but do not stand out relative to other changes of these proxies throughout the record. In contrast to Kaniewski et al. (2018), we do not see

indications for a long drought over several centuries, but rather two very rapid oscillations between an (intensely) wet and an (profoundly) dry state. These high amplitude fluctuations are followed by a period of drier conditions from ca. 3.8 to 3.5 ka. The two oscillations between 4.2 and 3.9 ka are consistent with proxy records from Italy and Algeria that suggest a double-peak centennial structure (Jalali et al., 2019). Similar to our observations, Schirrmacher et al. (2019) reported a dry phase from 4.4–4.3 $\pm$0.1 ka BP immediately followed by a shift to wetter conditions in two marine records from offshore southern Iberia.

The high-frequency isotopic change as observed in the Hermes Cave stalagmite at the 4.2 ka event also starts with a trend towards a severe drought (least negative $\delta^{18}$O values at 4.1 $\pm$ 0.2 ka), which is followed by a rapid shift towards very wet conditions (most negative value at 4.04-4.02 $\pm$ 0.3 ka). Afterwards, another slightly reduced dry-wet cycle follows until 3.85 ka. The maximum amplitude in the $\delta^{18}$O values corresponds to 1‰. If temperature variations are assumed to have a minor contribution, rainfall amount should be the dominating parameter. Speleothem and cave studies as well as modern rainfall

observations suggest a negative correlation between rainfall $\delta^{18}$O values and rainfall amount (more negative for higher rainfall amounts; Bar-Matthews et al., 2003; IAEA-WMO, 2019; Nehme et al., 2019). If temperature stays unchanged over the related period (and disequilibrium is non-existing or at least constant), it directly transfers into the calcite $\delta^{18}$O values. The water $\delta^{18}$O-rainfall amount sensitivity is about 1 ‰/290 mm in the Eastern Mediterranean (Fig. 3c; Supplementary data S1), comparable to observations at Soreq and Peqin Cave (Israel) with about 1‰/200 mm (Bar-Matthews et al., 2003). Thus, we expect a

relative rainfall variation of 15-30% (200-300 mm change relative to 1000-1300 mm annual precipitation) during the 4.2 ka event at Hermes Cave and potentially enhanced at lower elevation sites, which do not benefit from rain-out effects as it is the case for the Killini/Ziria mountain range.



## 5.3 Comparison with other regional records

In comparison with other records, we focus on the long-term trends and refrain from discussing high-frequency features below the centennial scale. A regional analysis on the Peloponnese is complemented by a larger spatial scale assessment that includes records from the Mediterranean, South-East Europe and the Alps.

$\delta^{13}$C values of southern Greek speleothems vary between -7 to -10 ‰ (Fig. 8), reflecting similar local conditions and vegetation. However, a detailed comparison of the $\delta^{13}$C values with other stalagmites from the Peloponnese and the surrounding region does not show coherent temporal signals (Fig.8), potentially due to different factors influencing the $\delta^{13}$C values at the corresponding sites. This is different for the $\delta^{18}$O values where similar trends can be observed throughout the Peloponnese and the Aegean (Fig. 9). The evolution of the $\delta^{18}$O values of Hermes Cave stalagmite GH17-05 can be separated

into three phases: (1) fluctuation around a mean value from ca. 4.6 ka to ~3.7 ka, (2) a trend towards less negative values from ca. 3.5 ka to ~2.0 ka indicating a drying trend and (3) a trend towards more negative $\delta^{18}$O values from ~2.0 ka to 0.8 ka, suggesting generally wetter conditions. A good agreement of the trends is found for other speleothems from the Peloponnese, e.g., the record from Mavri Trypa (Finné et al., 2017) that formed during three discrete growth periods overlapping with the time GH17-05 grew. The record from Mavri Trypa provides a similar climate picture with generally wetter conditions

suggested for the periods between 4.7 and 4.3 ka followed by rapidly oscillating $\delta^{18}$O values structured in a similar way as in the case of Hermes Cave and a growth hiatus in the drier period (least negative $\delta^{18}$O in GH17-05). In Mavri Trypa, wetter conditions between 3.8 and 3.5 ka are followed by a trend towards drier conditions culminating at 2.9 ka when growth ceased again. From 2.1 ka onwards, a trend towards wetter conditions in Mavri Trypa reflects the conditions as recorded in Hermes Cave. In addition, stalagmites from Alepotrypa Cave also show a trend towards less negative $\delta^{18}$O values between 5.0 ka and

3.0 ka and subsequently a trend to more negative values (Boyd, 2015). Furthermore, both records show a high degree of consistency in medium and high-frequency fluctuations.

Sediment cores from lakes complement the paleoclimate assessment. Lake Stymphalia in close vicinity to Hermes Cave has been strongly influenced by human activity in its catchment over the last 2500 years and shows only limited proxy variation prior to approximately 1820 cal BP (130 CE) (Seguin et al., 2019). Thus, a comparison of the trends with the Hermes

Cave record is difficult. However, some periods with enhanced erosion markers in the lake record (Rb/Sr) coincide with wetter periods or peaks in the speleothems $\delta^{18}$O record (1.8-1.4 ka; around 3 ka; 5.0-4.3 ka). Similarly, the Gialova $\delta$D record from the Western Peloponnese (Katrantiotis et al., 2018) does not agree exactly with the GH17-05 peaks, but matches e.g. at the wet period after 1.5 ka and the drier period from 2.8-1.5 ka. On the other hand, the trends in the $\delta$D values from Lake Lerna (Katrantiotis et al., 2019), about 50 km to the east of Hermes Cave, agree well within uncertainty with the $\delta^{18}$O record of

GH17-05.



On the wider regional scale, some agreement in trends is observed with the record from Lake Ohrid (Western Macedonia) showing consistent drying up to 2.0 ka with subsequent increase in wetness up to 1.7 ka (Lacey et al., 2015), where both records start to diverge. The low-resolution Lake Gölhisar record in Southern Turkey (Eastwood et al., 2006) and the high-resolution Jeita Cave record from Lebanon (Cheng et al., 2015) show consistency in the long-term trends with GH17-05. Kaniewski et al. (2019) noted colder and drier conditions from 3200 to 2900 BP for coastal Syria, which coincides with less negative $\delta^{18}O$ values in Hermes Cave from 3.0-2.8 ka. In Corchia Cave in Central Italy, a drying trend from 4000 to 2400 BP is observed with $\delta^{18}O$ values getting more positive (Isola et al., 2019). At least partially anti-correlated are climate records from Hungary with a trend towards more negative $\delta^{18}O$ values between 4000 and 3500 BP (Demeny et al., 2019) or from Lake Shkodra (Albania/Montenegro) with a trend towards more negative $\delta^{18}O$ values from 3500 and 2000 (Zanchetta et al., 2012). Partial correlation is visible with records from northern Turkey (Sofular Cave; Fleitmann et al., 2009), in particular, the drying trend from 3.5-1.6 ka ($\delta^{13}C$ in Sofular), and trend towards increased wetness from ca. 2.0 ka to 0.5 ka at Closani Cave in Romania (Warken, 2017). No significant correlation is apparent for northern Greece (Lake Dojran; Francke et al., 2013).

Beyond proxy information from other stalagmites from the Peloponnese and the closer region as well as data from lake sediment cores, marine sediment cores from the Ionian and Aegean Sea allow for additional comparisons. Sea surface temperatures (SST) in the Ionian Sea decreased from ~4.8 to ~2.7 ka by up to 6°C (Emeis et al., 2000). Temperatures recovered rapidly around ~2.5 ka and only marginally decreased afterwards until 1.0 ka. Changes in the Adriatic Sea SSTs are less pronounced (amplitude 2°C), but also show a minimum at ~ 2.9 ka (Sangiorno et al., 2003). Related to the lower SSTs, a decline in the warm-loving species in the Adriatic and Aegean Sea is observed. Minimum abundances of warm-species foraminifera were found in the Aegean Sea between 3.7 and 2.5 ka (Rohling et al., 2002). The decreasing trend in Ionian Sea and Adriatic Sea SSTs and the period of reduced warm-species foraminifera overlaps with the time period of increasingly positive $\delta^{18}O$ values in the Hermes Cave stalagmite, suggesting a direct climatic connection.

## 5.3. Implications

The climatic evolution in Southern Greece appears to be mainly modulated by the prevailing atmospheric circulation, in particular the North Sea/ Caspian Atmospheric Pattern and the North Atlantic Oscillation (e.g., Katrantsiotis et al., 2019). The climate of the Peloponnese often follows a similar pattern as seen in other Eastern Mediterranean archives (e.g., Finne et al., 2019), modulated, however, on a local scale mainly by topography. Based on the Hermes Cave stalagmite we summarize the main climatic changes on the Peloponnese from ca. 4.7 ka to 0.8 ka as follows:

- Two long-term trends are reflected in rainfall/infiltration and potentially to a minor degree in the temperature: an evolution towards drier (and potentially cooler) conditions from ca. 4.0 to 2.0 ka, followed by a trend towards wetter



conditions from 2.0 ka to 0.8 ka. Long-term changes in the $\delta^{18}O$ values (amplitude ~0.5‰) translate into rainfall variations of ca. 100 mm/a (assuming a similar $\delta^{18}O$/rainfall relationship as in Soreq Cave, Israel)

- The long-term trends are overlain by significant variations on the (bi-)centennial scale
- Within the interval 4.2-4.0 ka, an outstanding oscillation with the highest amplitude in the $\delta^{18}O$ values of the whole record with a duration of ca. 130 years occurred. Strong signals are also visible in Mg/Ca, $\delta^{13}C$, P/Ca, Ba/Ca.
- Until ca. 3.7 ka moderately wet conditions prevailed, interrupted by the oscillations at 4.2-3.9 ka BP when conditions rapidly shifted twice from drought to very wet conditions.
- Drier conditions related to the higher frequency fluctuations are inferred from Mg/Ca, $\delta^{13}C$, $\delta^{18}O$, and P/Ca at ca. 4.1, 3.9, 3.7-3.5, 3.4, 3.3, 2.8, 2.6, 2.0, and 1.4 ka

## 6. Conclusions

Stalagmite GH17-05 from Hermes Cave provides a new, continuous paleoclimate record for the Northern Peloponnese for the period from ca. 4.7 to 0.8 ka. The stalagmite growth period covers several well-known cultural periods and provides a climatic frame in which the societal changes can be discussed. Two long-term trends were identified: an evolution towards drier conditions and potentially lower temperatures from 3.7 – 2.0 ka, followed by a trend towards wetter conditions from 2.0 ka to 0.8 ka (end of stalagmite growth). The long-term trends are overlain by high-frequency fluctuations between dry and wet periods, which includes two drastic and rapid shifts (130-150 years duration) at 4.2-3.9 ka.

A comparison with other climate records from Greece and the surrounding seas indicates a good agreement regarding estimated trends in rainfall. The comparison provides highly important insights into regional changes and allows constraining major meteorological/climatic changes on the regional scale. Furthermore, the observed long-term changes in rainfall during the mid- to late Holocene of 10-15 % and up to 30 % on short-term multi-decadal scale at the high-elevation Hermes Cave site can provide constraints for assessing future challenges to the current water supply of the region. Most of the higher frequency climatic changes on the Peloponnese were found to occur on the centennial scale, demanding for critical evaluation of its influence on societal changes, i.e., how strong the impact of moderate changes on centennial scale is relative to slow changes on millennial scale. A special case are the high-amplitude shifts at 4.2 ka where a major shift occurred within 60-70 years and may therefore have had significant impact on society.

**Data availability**

Data is included in Tables 1 and 2 and additionally given in supplementary files (elemental ratios and isotope ratios vs. depth, clumped isotope raw data).



## Author contribution

I.U. developed the project idea, organized and led a field trip for reconnaissance and stalagmite sampling and contributed to data evaluation, interpretation and manuscript drafting, M.F. supported the reconnaissance field trip, actively sampled the stalagmite together with T.K. and equally contributed to data evaluation, interpretation and manuscript drafting, T.S.M. studied the stalagmite in detail, prepared samples for dating and isotope analysis, measured clumped isotopes, and provided fundamental input for manuscript drafting. E. E. assisted in stable isotope analysis and interpretation and actively contributed to the manuscript writing. R.M.-K. measured the elemental ratios of the stalagmite and helped in interpretation and manuscript drafting. D.S. helped constructing the chronology, supported the data interpretation and assisted in manuscript preparation. N. F. contributed to the Th-U chronology and helped drafting the manuscript. T.K. took part in the field trip and the stalagmite sampling, supervised the data acquisition, and drafted the manuscript.

## Competing interests

The authors declare that they have no conflict of interest.

## Acknowledgements

We thank the Ephorate of Palaeoanthropology and Speleology of Southern Greece for permitting visits to and sampling in Hermes Cave. We thank Chryssia Kontaxi and Dimitris Karoutis for introducing us to the cave and Linn Haking for support in survey, sampling and documentation. Field work for TK, coring equipment and isotope analyses were financed by the Heidelberg Graduate School of Fundamental Physics (HGSFP). Regarding clumped isotope analyses, we acknowledge the technical help of the team 'physics of environmental archives' to maintain the IRMS instrument that was funded through the grant DFG-INST 35/1270-1 and are grateful to Henrik Eckhardt for implementing a customized data evaluation program. M.F. acknowledges support by the Swedish Research Council (VR; grant number 421-2014-1181). DS acknowledges funding from the DFG (SCHO 1274/11-1). We thank Carla Roesch and Sandra Rybakiewicz for Th/U preparation, René Eichstädter for MC-ICPMS measurements and quality control, and Sophie Warken for helpful suggestions regarding the Th/U chronology.

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



# Tables

**Table 1**: Results of radiometric analysis of calcite from GH17-05. $(^{230}Th/^{238}U)$ and $(^{230}Th/^{232}Th)$ refer to activity ratios. All measurements are reported with $\pm 2\sigma$ uncertainties. Corrected ages are given relative to a detrital correction model with an initial $(^{230}Th/^{232}Th)$ activity ratio of $1.92 \pm 0.96$ of the contaminating phase.

| Depth | Analysis | $^{238}U$ | $^{232}Th$ | $(^{230}Th/^{238}U)$ | $(^{230}Th/^{232}Th)$ | $\delta^{234}U_{initial}$ | Uncorr. Age b2k | Corrected Age b2k |
|---|---|---|---|---|---|---|---|---|
| (mm) | ID | (ng/g) | (ng/g) | act. ratio | act. ratio | (‰) | (ka) | (ka) |
| 4 | 9363 | 179.83±0.01 | 11.275±0.023 | 0.0412±0.0004 | 2.02±0.02 | 58±2 | 4.31±0.04 | 0.21±1.24 |
| 7 | 10248 | 183.89±0.02 | 3.410±0.007 | 0.0209±0.0003 | 3.45±0.06 | 56±3 | 2.16±0.04 | 0.97±0.59 |
| 10 | 9640 | 224.70±0.01 | 3.476±0.006 | 0.0275±0.0002 | 5.44±0.04 | 50±1 | 2.88±0.02 | 1.88±0.52 |
| 17.5 | 9364 | 198.78±0.01 | 1.236±0.003 | 0.0301±0.0003 | 14.86±0.14 | 37±2 | 3.20±0.03 | 2.81±0.21 |
| 22 | 10249 | 183.53±0.02 | 1.557±0.002 | 0.0326±0.0003 | 11.74±0.11 | 44±2 | 3.44±0.03 | 2.90±0.27 |
| 26 | 9763 | 160.20±0.01 | 1.132±0.004 | 0.0389±0.0004 | 16.91±0.18 | 39±2 | 4.15±0.04 | 3.70±0.23 |
| 33 | 9764 | 183.58±0.01 | 0.725±0.004 | 0.0423±0.0005 | 32.90±0.42 | 46±2 | 4.49±0.05 | 4.25±0.13 |
| 36 | 10250 | 209.86±0.02 | 0.3733±0.0006 | 0.0411±0.0003 | 70.55±0.55 | 45±2 | 4.36±0.04 | 4.27±0.07 |
| 40 | 10251 | 155.99±0.02 | 1.384±0.002 | 0.0455±0.0005 | 15.61±0.18 | 47±4 | 4.83±0.06 | 4.27±0.29 |
| 42 | 9356 | 168.52±0.01 | 3.229±0.009 | 0.0518±0.0004 | 8.30±0.06 | 46±2 | 5.53±0.04 | 4.29±0.62 |

**Table 2**: Results of clumped isotope analysis of selected key sections of stalagmite GH17-05. n = number of replicates. The uncertainty of the $\Delta_{47}$, $\delta^{13}C$ and $\delta^{18}O$ values are given as standard deviation, for the temperature based on the standard error. $\delta^{18}O_{dripwater}$ is a calculated value based on calcite $\delta^{18}O$ and $T_{\Delta47}$. The uncertainty of the calculated $\delta^{18}O_{dripwater}$ includes the uncertainty in calcite $\delta^{18}O$ and $T_{\Delta47}$ *standard deviation of reference carbonates (reproducibility).

| Depth | $\Delta_{47}$ | $T_{\Delta47}$ | n | $\delta^{13}C$ | $\delta^{18}O_{calcite}$ | $\delta^{18}O_{dripwater}$ | Age b2k |
|---|---|---|---|---|---|---|---|
| (mm) | (‰) | (°C) | (-) | (‰) | (‰) | (‰) | (ka) |
| 4 | 0.752±0.010 | 7.6±2.3 | 2 | -7.83±0.21 | -7.48±0.07 | -9.1±0.5 | 1.1±0.9 |
| 9 | 0.741±0.020 | 10.5±6.8 | 2 | -9.55±0.02 | -7.46±0.06 | -8.4±1.4 | 1.7±0.9 |
| 14 | 0.755±0.030 | 6.6±6.9 | 3 | -9.26±0.05 | -7.19±0.08 | -8.9±1.4 | 2.4±0.5 |
| 16 | 0.748±0.003 | 8.5±1.2 | 3 | -9.74±0.03 | -7.26±0.02 | -8.6±0.3 | 2.6±0.5 |
| 21 | 0.737±0.007 | 11.8±2.0 | 2 | -9.35±0.04 | -7.28±0.04 | -7.9±0.4 | 3.1±0.4 |
| 27 | 0.752±0.009 | 7.5±3.3 | 2 | -9.36±0.01 | -7.53±0.01 | -9.1±0.7 | 3.7±0.3 |
| 32 | 0.728±0.030 | 14.3±6.9 | 3 | -9.51±0.14 | -7.31±0.14 | -7.3±1.4 | 4.1±0.2 |
| 37 | 0.775±0.020* | 1.4±6 | 2 | -9.39±0.03 | -7.51±0.09 | -10.4±1.2 | 4.40±0.2 |





**Figures**

A:

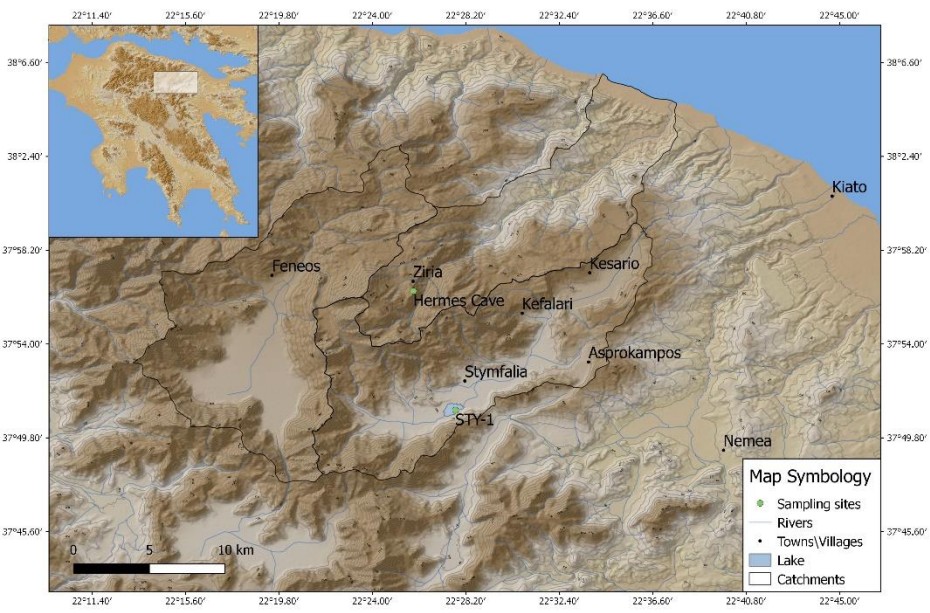

B:

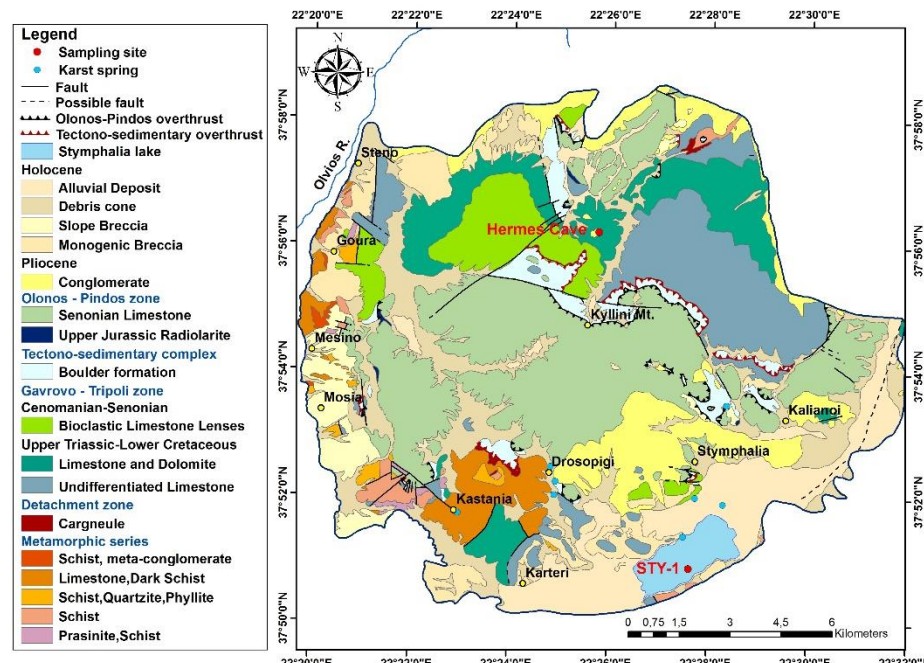

**Figure 1: (A)** Map of the study area in north-eastern Peloponnese **(B)** Geological Map of the study area (modified after Nanou and Zagana, 2018). Hermes Cave is located at the centre of the Ziria Massif at about 10 km from Lake Stymphalia.





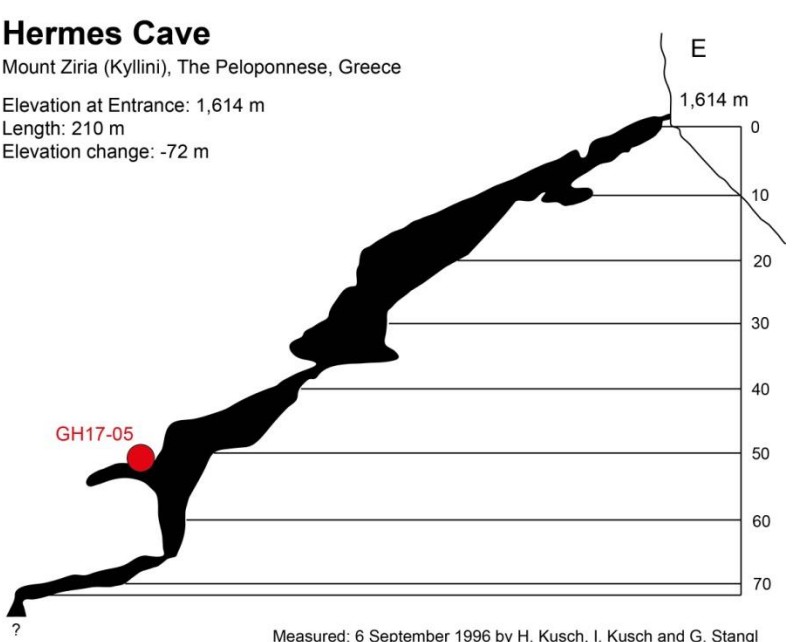

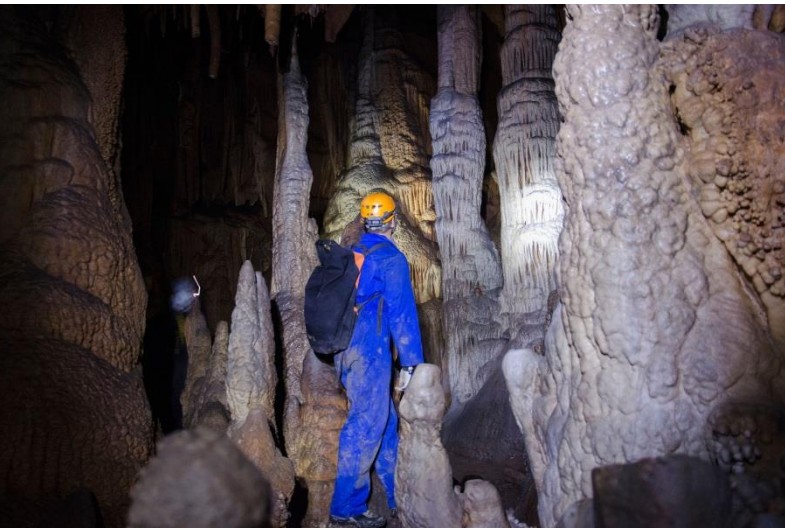

**Figure 2:** Profile plan of Hermes cave at the Ziria Massif (Fig. 1) with sampling position (A) and inside view (B).



A:

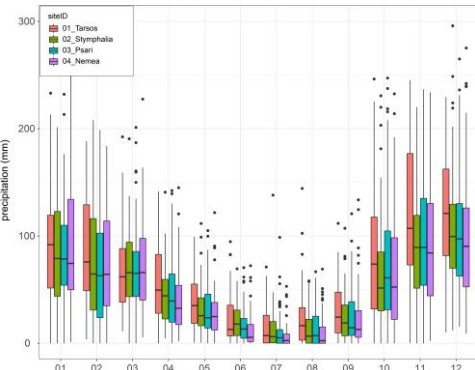

B:

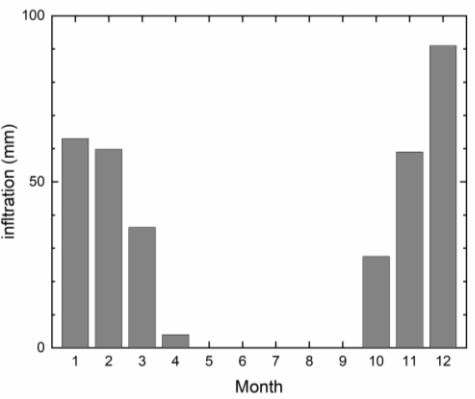

5   C:

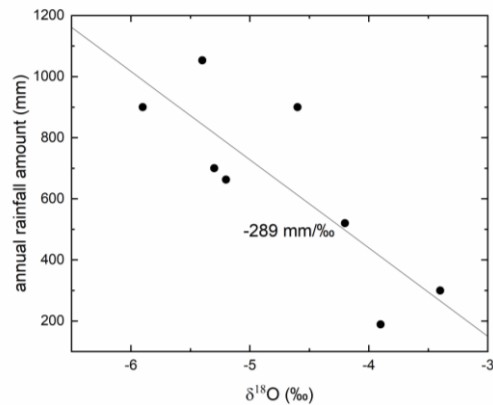

**Figure 3:** A: Monthly precipitation amounts at four meteorological stations of the Greek Special Secretariat for Water (YPEKA) around Mt. Ziria with the 25-percentile range (boxes) and single outliers (dots), measured from 1950-2010 (Tarsos since 1964). Tarsos in the west receives significantly more precipitation than Nemea in the east of Ziria. B: Typical infiltration pattern of Southern Greece, shown at the

10    example of Athens (IAEA-WMO, 2019). Note that between April and October typically no infiltration occurs. C: Sensitivity of the annual average rainfall $\delta^{18}O$ value on annual rainfall amounts in the Eastern Mediterranean. The analysis is based on data from the IAEA GNIP stations (IAEA-WMO, 2019) and published values (Nehme et al., 2019).




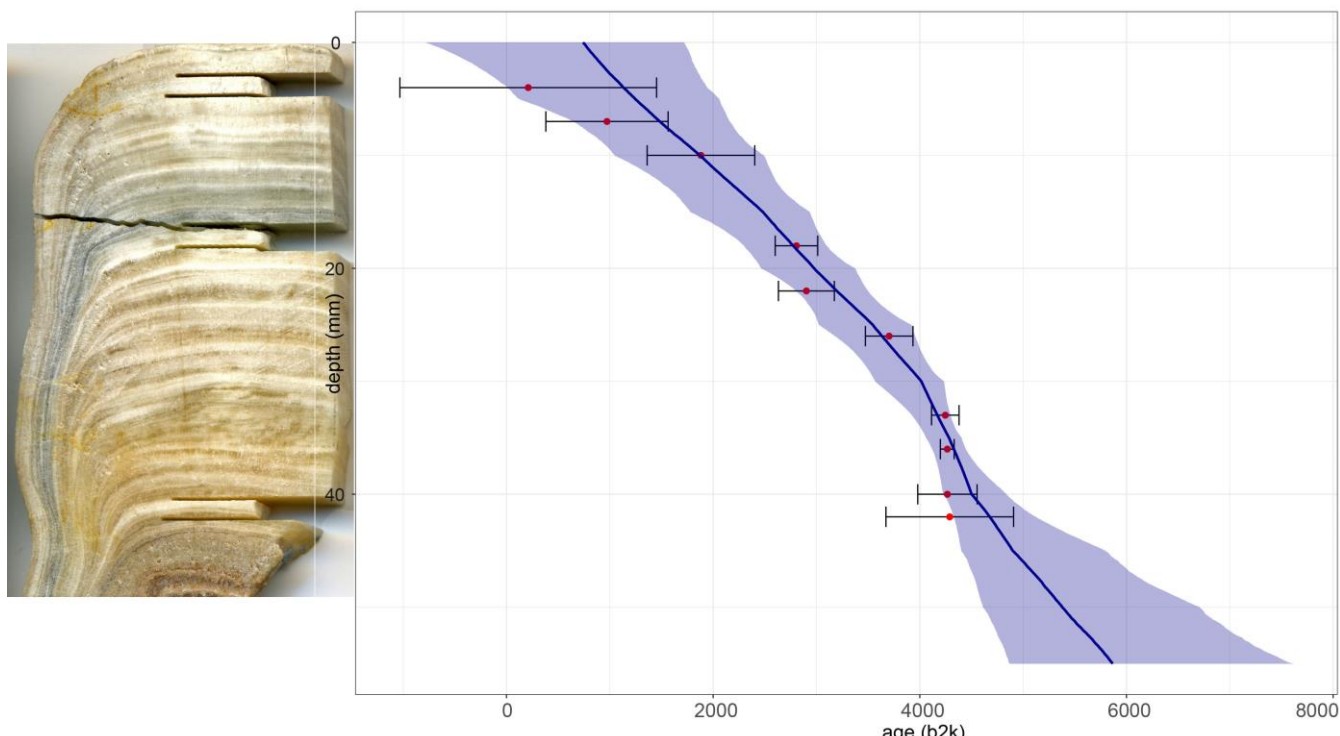

**Figure 4:** Slab of the stalagmite in reference to the chronology based on Bayesian age-depth-modelling using the R package Rbacon (v.2.3; Blaauw and Christen, 2011).





**Figure 5:** Stacked graph of stalagmite GH17-05 δ¹⁸O and δ¹³C values as well as Mg/Ca and P/Ca ratios. Note that higher Mg/Ca ratios point downwards. The blue shaded time periods are related to the 4.2 ka (section 5.2) and the 2.8 ka phase.





**Figure 6**: Elemental ratios of stalagmite GH17-05 vs. age. The blue shaded time periods are related to the 4.2 ka and the 2.8 ka phase.







**Figure 7:** Evolution of the δ$^{18}$O values of Hermes Cave stalagmite GH 17-05 between 4600 and 3600 years b2k.





**Figure 8:** Stalagmite $\delta^{13}C$ values from the Peloponnese (Kapsia, Hermes, Mavri Trypa), Northern Greece (Skala Marion) and the wider Eastern Mediterranean region (Sofular Cave, Turkey, Closani Cave, Romania). More negative $\delta^{13}C$ values are generally interpreted to reflect wetter conditions coinciding with more active vegetation. This does not hold for all time periods at Hermes Cave (see section 5.1). The blue shaded time periods are related to the 4.2 ka and the 2.8 ka phase.





**Figure 9:** Stalagmite δ18O values from the Peloponnese (Kapsia, Hermes, Mavri Trypa) and Northern Greece (Skala Marion) and the wider Eastern Mediterranean and Eastern European region (Sofular Cave, Turkey, Closani Cave, Romania). In the case of Sofular Cave, δ13C values have been interpreted to reflect hydrological changes rather than the δ18O values. The blue shaded time periods are related to the 4.2 ka and the 2.8 ka phase.