# Peer review of "A 4000-year long Late Holocene climate record from Hermes Cave (Peloponnese, Greece)"

_Climate of the Past, 2020_

## Referee Comment (RC1) · Anonymous Referee #1 · 13 Jul 2020

Reviewer comment for A 4000-year long Late Holocene climate record from Hermes Cave (Peloponnese, Greece), from Tobias Kluge et al, submitted to Climate of The Past The ms from Kluge et al presents a multiproxy record (stable isotope, clumped isotopes, trace element composition) from a speleothem from Hermes cave, in the NE Peloponnese (Greece). The speleothem growth interval spans from ca. 4.5 ka to the present (though the top age is affected by large uncertainty) and thus covers a period of particular archaeological interest for the region (i.e. the late Bronze Age-Iron Age). Due to large uncertainty in the calibration, clumped isotope results are mostly used to infer equilibrium deposition. The $\delta$18O is interpreted in terms of hydrological changes (i.e. amount effect), according to previous works from the region, whereas others proxies are tentatively addressed as related to changes in hydrology, infiltration and soil conditions. From the multiproxy record, authors identify a long-term trend of drying from ca. 2 to 2 ka and two events of drier conditions around 4.2 and 3.2 ka. Through comparison with others archives from the region, the authors placed the observed variability in the broader palaeoclimatic framework available for the Eastern Mediterranean. The paper is properly written and structured, and the analytical methods appear robust. However, I found the proxy interpretation very forced in some points, and not fully supported by the data. This often led to an over-interpretation of the record and to a rather speculative discussion. Also, though I appreciated the effort in trying to quantify the observed variations, I found the attempt over-simplistic, because a proper monitoring is lacking and the analyses of present-day conditions is not robust enough. Overall, in its present form the paper cannot be accepted- It should be potentially published in Climate of the Past, but only after major revision and consistent rethinking and rewriting. Please find specific comments below. Major points: Petrography: Petrographical information are totally lacking. In my opinion, they are mandatory for any speleothem research (they are also very cheap and easy to achieve!). The occurrence of specific calcite fabric (columnar) is considered the best way to assess deposition close to isotopic equilibrium (see e.g. Frisia et al., 2000; Frisia and Borsato, 2010; Frisia, 2015; Faichild and Baker, 2012...), whereas other fabrics were demonstrated to be affected by large kinetic effects. I strongly suggest to prepare at least one thin section (as the fabric seems quite constant) and to report proper information about it. Age modelling I do not agree with the choice of the age model: looking at Fig. S5 I found the stal-age model more convincing, wheras the bacon one shift both the lower and the upper part of the record to unrealistic older ages. I agree that it doesn't affect too much the age of the interval discussed in details, but I think the stal age one is more correct and should be used instead Equilibrium deposition and $\delta$18O interpretation The assessment of equilibrium and the determination of dripwater $\delta$18O through clumped isotopes is an interesting approach, novel but already supported by previous studies. However, its use in this work relies on just one actual temperature measurement, which is not enough nor totally significant in my opinion for quantification purposes, especially in shallow environments like this, which are likely affected by strong seasonal differences in ventilation and temperature (i.e. only 55 m from the entrance). This make the discussion in p. 8 lines 1-11 and 26-28 and p 9 lines 1-6 and 8-24 rather speculative. Also, though I agree that clumped isotope results suggest a low degree of kinetic fractionation and that thus the speleothem $\delta$18O can be a proxy for the $\delta$18O of the precipitation, the calculation of drip water $\delta$18O based on the modern temperature is not robust enough to be presented. I agree that rainfall amount can be the main driver of rainfall $\delta$18O (and thus of speleothem) in this setting, as already shown by a number of studies and by precipitation monitoring in the region. However, this is not the only driver: changes in the seasonality would affect the final $\delta$18O and are very difficult to quantify (as correctly stated), but also changes in the source of moisture can be important, though tricky to detect. As example, in mountain regions during summer there is usually a large proportion of moisture due to local evaporation and convective precipitation, whose isotopic composition cannot be simply related to amount effect. Also, during winter, the southern Balkans are interested by incursions of cold air from NW Asia, which correlated to increased snow cover, likely influencing the annual budget and thus the mean annual value of recharge. And there are many others variables... The influences of these effects are often difficult to disentangle and may be contrasting or may change during time (e.g. Dragusin et al., 2014). This means that a simple quantification of $\delta$18O in terms of changes in the amount of rainfall is not straightforward and should be avoided in absence of a proper long-term monitoring of rainfall $\delta$18O values at the specific cave locations. Please add more discussion about other potential effects and remove the quantification attempt (e.g. p4 lines 4-5, pS1 lines 23-26, p 9 lines 15-16, p. 12 lines 23-32) Interpretation of $\delta$13C I found this part rather problematic. Honestly, I'm not able to see any common trend in curves presented in Fig. 5. $\delta$13C and $\delta$18O are not consistently anticorrelated as the paper claims, and to me their variations are largely disconnected. Also the proposed explanation for the supposed anticorrelation is not convincing at all, as $\delta$13C values always remain in a range which is consistent with biogenic input from soil (see e.g.

Tremaine et al., 2011), moreover, the few examples of anti-correlation related to short infiltration times (Bar Matthews et al., 2003 but also Regattieri et al., 2018) consist in sharp opposite peaks in specific and restricted intervals of the records, and not to a slightly contrasting pattern in some points. I think the main drivers of $\delta$13C are changes in the biogenic CO2 input and thus can be related to vegetation and soil state. The fact that they are not consistent with $\delta$18O likely means that hydrological changes were not strong enough to deeply affect the soil state. Trace element interpretation Also this part is quite problematic. Again, I do not see any clear common trend between $\delta$13C and P/Ca or between $\delta$18O and Mg, and the interpretation of the P/Ca and Mg/Ca record is a bit odd and simplistic. P in speleothem has several potential origins, depending on individual cave settings. For example, where P-rich minerals like apatite are present, it can be sourced from the bedrock and be incorporated according to a distribution co-efficient in the crystal lattice as P-rich phases (Frisia et al., 2012). However, apatite is very rare. Instead, several works from temperate ecosystems indicate that phosphorus in cave drip water derives principally from the leaching of decomposing plant residue (Borsato et al., 2007; Treble et al., 2003). Thus, P concentration is often interpreted as proxy for infiltration and/or as a surface bioproductivity marker (Fairchild and Treble, 2009), with variations related to changes in vegetation and soil conditions. The fact that here its variation are decoupled from those of the $\delta$13C likely suggest the absence of major environmental variations throughout the studied period. Also the interpretation of Mg is not convincing. Due to its high solubility, it is mostly transported as solutes (Fairchild and Treble, 2009) and not within mineral detrital particle. A certain flux as solid is possible, but in this case I would aspect a stronger correlation with Al and Mn, whose variations instead are largely disconnected from that of Mg/Ca. The only thing that I see from the Mg record is that there is a slight similarity in its long term trend with that of the $\delta$18O (both increasing), which may be due to hydrological variations (increased residence time during drier condition). U isotopes Also here there are some problems in my opinion. For what I know, variations in speleothem [234U/238U]i can be related to changes in the relative proportion of U deriving from the carbonate bedrock

versus that originating in soil (Kaufman et al., 1998; Ayalon et al., 1999; Frumkin and Stein, 2003; Hercman et al., 2020). Water interacting with more developed soil should have a higher 234U content, due to greater surface of mineral-water interaction. Thus I would aspect to have higher 234U/238U when the soil is more developed, i.e. under wetter conditions. Again, to me it points to a lack of major variations in soil and vegetation status during the studied period. 4.2 event: I agree that the oxygen record shows a hydrological change at that time (though not very prominent), but to me the other proxies do not, implying a very subdued expression at the cave site and a not very strong environmental response. Comparison with other records: As correctly stated, the comparison between $\delta$13C record from the Peloponnese does not shows any common trend, so in my opinion figure 8 could be avoided. I also found the correlation with others $\delta$18O speleothem records a bit forced. I acknowledge some similarity in the general trend of some of the curves in fig. 9, but it is really really weak. Authors should be more honest in recognizing that most of the variations is not totally replicated, especially at the multicentennial scale. Also, why compare with the $\delta$18O record from Sofular cave? This record mostly records changes in the proportion o moisture from the Med or the Black see, and it is not related to variations in the amount of rainfall!! The whole discussion from p 13 line 22 to p 14 line 22 about correlations with lake records etc make no sense because it is not supported by a figure. The reader must evaluate by itself the correlation among the records. The whole paragraph is advertising, not science…. Other points: p1 line 22: change elemental ratios to trace element composition of to elemental/Ca ratios p1 line 25: high degree of correlation is a bit too strong, I would say similarity p2 line 11: Add However before A paleoclimate p2 line 13: This sentence reads a bit odd, please rephrase like "Here we focus on a speleothem from Hermes Cave (NE Peloponnese, Greece) and compare our record with others climatic archivers, notably speleothems and lake sediments, from the same region" or similar p2 line 23:Add a proper ref after conditions. I would also add that temperature quantification in stalagmite is often complex p2 line 28-30 I would delete the sentence about the climate divide in the Peloponnese, as this point is not further

discussed. p 3 line 13: Is the vegetation totally composed of C3 plants? p3 line 25-31 it is not clear here or in the supplementary how the sensitivity of infiltration to temperature is calculated p3 line 32: Regarding infiltration as snow, I'm not very familiar with the specific cave setting, but I guess that most of the recharge is rain and not snow and that winter snow only lasts shortly, not enough to stratify in layers equilibrated and not equilibrated with the atmosphere. p4 line 7-12: Petrographical information must be added here. Also , the soot layer is mentioned here and in the abstract, but it is not further discussed in the following sections. p 6 line 25-32 I appreciate this approach and this discussion, however there is a repetition with lines 25-31 of page 4, please collate the information here or in the methods only. p7 lines 11 and 15: add the proper $\pm$ symbol. p7 Elemental ratios: Are they molar ratios or simply element/Ca ratio with the Ca values considered invariant? Is not clear. Please explain and be consistent (and I would prefer just the element expressed in ppm and discussed as concentration, as the ratios are mostly used for elements in solution like Mg, Sr, Ba to discuss the occurrence of PCP. It would be interesting to see the value expressed as ppm, as in my experience very often elements such as Pb and Mn are very low in concentration and their changes not very significant. p7 line 26-27: I see just a slight similarity in the Al, Mn and Fe curve, not a clear correlation. A correlation table reporting r values between each pair of elements have to be inserted, and also, it would be better to plot the ratios discussed together one close to each other (see comments on figures). p7 28-29: Given that I do not see any correlation between P, $\delta$13C and $\delta$18O, this information about the same number of peaks is not very useful (and also a bit misleading). p7 29-30 The correlation between elevated Ba and Sr and Pb and Mn is very hard to evaluate from Fig. 5 p7 32: Can you quantify the shift and the associated chronological mismatch between the isotope and t.e. records? p8 line 2: Add the standard deviation between samples p 11 line 4-5: if Mg variations are related to hydrological variations, it makes no sense to calculate a related temperature change. Please remove.

Figures:

[Figure]

In general they are not of very good quality. Please enlarge the y axis and make the blue bands lighter, as they actually cover the curves and make difficult to evaluate them.

---

## Referee Comment (RC2) · Anonymous Referee #2 · 13 Jul 2020

Klunge et al present a new multiproxy speleothem record from Peloponnese ranging from ca. 0.8 to 4.7 ka with the aims to give new data on climate evolution to place societal and cultural evolution of this important part of the Mediterranean during Bronze and Iron age. Despite this ambitious target, the manuscript basically fails to obtain important insight on it, because the high clastic contamination produce a large spread of U/Th ages and associated error, and the signal of many proxies is not always pronounced. The authors are aware of this and at certain points of the manuscript tried to focus on a short interval close or corresponding to the so called 4.2 event. Also in this case the discussion is not able to focus on substantial new ideas on this period. So the general aspect of the manuscript is confused in the treatment of the data in the discussion and then in the abstract and the conclusion. First of all the manuscript needs a

complete reorganization in the aims and introduction. The present aims are good for a speleothems with better chronology and much better signal. Along the manuscript, we pass in a confounding way (some time with repetitions) between the description of the long-term record and the short term changes. These are some of the general comment I have. However, there are many points along the manuscript, which needs to be improved. I try to give some example below. Abstract Pag. 1 Line 15 the 3.2 even is mentioned, and in many figures is highlighted but along the text I never see a serious discussion on that. Moreover, the chronology of this interval is really poor. Pag. 1 Line 20 the record is reported continuous between 800 and 5300 ka differently from the conclusion Pag. 1 line 24 234U/238U: there is no particular discussion on this point along the text to be so relevant to be mentioned in the abstract (indeed is lacking in the conclusion). Introduction Pa.g 2 lines 14-15. I think we must be aware that speleothem can be precisely dated if clastic contamination is negligible. Pag. 2 lines 30-31 This the style of the manuscript a description of a short event and then a focus on long term trend. So, most of the introduction is not useful to justify this view.

Study Area Pag. 4 lines 2-5 The data from Nehme et al., 2019 and Bar-Matthews et al., 2003 cannot be used acritically for Peloponnese. They cannot be presented as valid data for your area. There is a general "paradigmatic" view on the interpretation of the d18O in the Mediterranean and I can agree this can be used for past reconstruction. Moreover, in your discussion you try to justify this view using also other proxies. I think this is a correct "qualitative" approach. In absence of regional-to-local convincing data on precipitation to show data from other sector is not good.

Material and Method

Pag. 4 line 10 "a soot layer…...can you please show the position in the figure 4. In the text there is no mention of thin section and just a brief description on the fabric would be useful also for the equilibrium conditions and to discuss if there are hiatuses. In some points it seems likely. Pag. 4 Line 17 "…where manually pre-treated to obtain pure carbonate…..." please can you explain more precisely?

Pag. 4 line 25. Clastic contamination can be also related to clastic-carbonate? I'm not an expert on U/Th measurements.

Results Pag. 6 line 24 "may be"? "is" better.

Pag. 7 lines 5-10. It Is unclear why you use two different Bayesian program and then choose one instead the other. Can you show both?

Pag. 7 line 14. "suggest" I understand what the authors want to say, considering the large error, but I prefer "indicate".

Pag. 7 line 19. Can you show this trend with a polynomial curve? Can you be statistically confident that this a trend or is just a visual impression? Please can you explain why in figure a different averaging is choose for Skala Marion. Can you show with a polynomial curve the trend described in the text of the Hermes Cave?

Pag. 8 Lines 5-8 "...the general correspondence of individual and average….." Considering there is only one single measure of cave temperature and the large T variability obtained using clumped isotopes and the associated error the conclusion would be: there is no secure conclusion. Discussion Pag. 8 lines 14-15. If the manuscript is focused on this interval this should declared since the introduction and the manuscript structure should be mostly different and most focused. But the general organization of the manuscript is not well done. There is no a clear focus. At the end what do the authors want solve? What do they have then solved? Pag. 8 lines 15-17. These two sentences are rather confounding. The chronological uncertainties are elevate for most of the record and not just on top. A detailed correlation with historical event is honestly not applicable (if we can exclude a brief interval). Indeed the second sentence is correct.

Pag. 8 lines 27-28. Once again I don't think to stress to this point is useful.

Pag. 8 lines 28-31. I don't think that the conclusion of Borsato et al. (2016) can be transported acritically from Alps to Peloponnese in a so strict sense without a general

monitoring program like the data presented by Borsato et al.

Pag. 9 lines 4-6. The ranges of values is quite large. In absence of more detailed local data many calculation are probably misleading.

Pag. 9 lines 15-16 The relationship reported by Bar-Matthews is very local, and it cannot used for Pelopennese.

Pag. 9 lines 20-24 this is one of the few point where other short term oscillation are considered.

Section 5.2 There are a lot of literature on 4.2 event, but the discussion proposed did not add relevant points. Line 28 Rousseau et al., 2019 is not a paper. So difficult to quote.

Pag. 12 lines 23-27 here there are some sentences and concept repetitions. Once again, I think is quite misleading to use the correlation defined for far areas.

Pag. 14, lines 14-16. It is hard to say that Mavri Trypa provides a similar climate picture.

Pag. 14, lines 20-21. "Furthermore, both records show a high degree of consistency in medium and hig-frequency fluctuation." Absolutely not. This is an overexploitation of the data. Moreover, there is a mention to high-frequency oscillations which have not been discussed in detail along the manuscript.

Pag. 14 lines 22-30. In some part of the manuscript the correlation with Lake Stymphalia is presented as strategic for the general interpretation. There are no proxy records show for this lake and the "comparision of the trends….is difficul". There are other lakes cited but the record are not shown. This section seems quite useless.

Pag. 14, lines 1-13 There is a discussion of records which are not show in any figure, so the comparison is difficult.

5.3 implications

Pag. 14 lines 23-24. This point appears here for the first time and there is no any discussion. This section is not "implication" but already a summary of the main result, some not discussed at all, like the list of drier events reported as last point at pag. 15.

Conclusion

Pag. 14 line 14. Where along the manuscript do emerge that there is a cooling trend? The introduction promise some conclusion related to social evolution and climate, but then?

Overall, I consider the manuscript not suitable for publication even if the data can have some interest. I suggest to change the target of the manuscript, basically deciding which is the main focus and what wants to solve really and not what would be interesting to solve. The chronology is relatively poor so, an honest and calibrated manuscript is necessary and in this case, for me, welcome.

---

## Referee Comment (RC3) · Anonymous Referee #3 · 18 Jul 2020

The manuscript presents a 4000-long palaeoclimate record from Late Holocene, from Hermes Cave (Peloponnese, Greece). The area and the period is obviously very interesting for archaeological, geoarchaeological and societal sciences too. During the last years/last decade, there have been very interesting, robust and solid studies of this period and in the general area of East Med, Aegean and Greece in particular. The level of current knowledge is quite high and any new addition should step on this and make a step forward. The major problem of the study by Klunge et al. is that their proxies, together with their assessment of the data, are not supporting their conclusions. It seems that the authors tried to use their available data and forces some conclusions which cannot be supported. The manuscript is well written and the methods used are indeed enough to proceed to a palaeoclimate study, but the results are not helping. It is

probable that the specific speleothem archive is actually not suitable for such a study.

Major remarks: - Age uncertainty. The stalagmite growth period is very short, so the age model must be very precise and accurate. Unfortunately, the GH17-05 calcite was not 'clean' enough to give proper U/Th datings for this purpose. It is impossible to proceed to such detailed and specific interpretation, discussion and reconstruction with such uncertainties in Late Holocene.

- Identification of phases. The authors proceed to identifying phases of wet and dry pulses, based on vague observations on the d18O curve. These observations must be justified, by any means of analysis, considering value trends, statistical evaluation (is a pulse an outlier from the rest of the curve?), normalization or any other way of analysis. In some proxies, a periodicity is given (eg. line 28, page 9) without any analytical calculation (naked eye?).

- Discussion. Correlation with other records is not properly justified. The observations by the authors are not really visible in the plots, even by comparing some of the records and excluding others. One needs to keep in mind the age uncertainties as well, in order to try to find correlating pulses between records. Eg. detecting dry conditions in time windows of 0.1 ka (eg. line 8, page 15) is not consistent with the age model.

- Figures. Figures need reorganization and improvement. Fig. 1 A, does not give a clear location of the cave. Fig. 1 B, is not really needed in such details. Fig. 2 B, photo is not helping actually. Fig. 4, the stalagmite needs more info plotted, such as the axis of sampled positions. Fig. 8 and 9, the references of the presented records are missing, the should be added here and in the reference list.

Generally, the article cannot be accepted in its present form. It needs an overall major revision and additional analytical effort (eg. age model) in order to be considered for publication in CoP.

---

## Author Comment (AC1) · 21 Sep 2020

Reply to Anonymous Referee #1

In the following we reply to the main point of criticism. Minor aspects (e.g., related to language and grammar) will be directly considered in a revised version. Referee comments are in italic and grey, the author response in black.

*The ms from Kluge et al presents a multiproxy record (stable isotope, clumped isotopes, trace element composition) from a speleothem from Hermes cave, in the NE Peloponnese (Greece). The speleothem growth interval spans from ca. 4.5 ka to the present (though the top age is affected by large uncertainty) and thus covers a period of particular archaeological interest for the region (i.e. the late Bronze Age-Iron Age). Due to large uncertainty in the calibration, clumped isotope results are mostly used to infer equilibrium deposition. The δ18O is interpreted in terms of hydrological changes (i.e. amount effect), according to previous works from the region, whereas others proxies are tentatively addressed as related to changes in hydrology, infiltration and soil conditions. From the multiproxy record, authors identify a long-term trend of drying from ca. 2 to 2 ka and two events of drier conditions around 4.2 and 3.2ka. Through comparison with others archives from the region, the authors placed the observed variability in the broader palaeoclimatic framework available for the Eastern Mediterranean. The paper is properly written and structured, and the analytical methods appear robust.*
- We are pleased by the general positive view of our study

*However, I found the proxy interpretation very forced in some points, and not fully supported by the data. This often led to an over-interpretation of the record and to a rather speculative discussion. Also, though I appreciated the effort in trying to quantify the observed variations, I found the attempt over-simplistic, because a proper monitoring is lacking and the analyses of present-day conditions is not robust enough. Overall, in its present form the paper cannot be accepted. It should be potentially published in Climate of the Past, but only after major revision and consistent rethinking and rewriting. Please find specific comments below.*
- A long-term monitoring does not exist, however, basic data are available from other studies (e.g., Kusch, 2000). In September 1996, H. Kusch measured a temperature of 7.6°C in the cave interior, slightly below the values measured during our cave visit in September 2017 of 8.95 and 9.2 °C at mark 19 and the deepest cave part, respectively. A large fraction of the higher temperature in 2017 may be explained by the warming global climate in the last two decades. Only the top-most part of the cave is relatively shallow (approximately 30 m below the surface). The cave levels dip with about 28-30° downwards (Kusch, 1996) with the deepest currently reachable cave part at 72 m below the entrance. Thus, the deepest parts are about 100 m below the surface. The small cave entrance at Hermes cave of only ca. 2 x1 m only allows restricted air flow. Based on the high thermal inertia of the hostrock, the cave temperature is highly unlikely to change beyond the variability known from similar cave systems (e.g., Bunker Cave; Riechelmann et al., 2011; Katerloch; Boch et al. 2010). We'll amend the manuscript with the corresponding discussion and the extra information.

*Major points:*
**Petrography:** *Petrographical information are totally lacking. In my opinion, they are mandatory for any speleothem research (they are also very cheap and easy to achieve!). The occurrence of specific calcite fabric (columnar) is considered the best way to assess deposition close to isotopic equilibrium (see e.g. Frisia et al., 2000; Frisia and Borsato,2010; Frisia, 2015; Faichild and Baker, 2012...), whereas other fabrics were demonstrated to be affected by large kinetic effects. I strongly suggest to prepare at*

*leastone thin section (as the fabric seems quite constant) and to report proper information about it.*

- A thin section has been prepared and investigated. The manuscript will be amended by the petrographic information.

***Age modelling:*** *I do not agree with the choice of the age model: looking at Fig. S5 I found the stal-age model more convincing, wheras the bacon one shift both the lower and the upper part of the record to unrealistic older ages. I agree that it doesn't affect too much the age of the interval discussed in details, but I think the stalage one is more correct and should be used instead.*

- Together with additional dating attempts (anthropogenic markers at the stalagmite top, e.g., C-14 bomb peak; further U-Th data for the rest and in particular the oldest stalagmite part) we'll carefully revisit the age modelling

***Equilibrium deposition and δ18O interpretation:*** *The assessment of equilibrium and the determination of dripwater δ18O through clumped isotopes is an interesting approach, novel but already supported by previous studies. However, its use in this work relies on just one actual temperature measurement, which is not enough nor totally significant in my opinion for quantification purposes, especially in shallow environments like this, which are likely affected by strong seasonal differences in ventilation and temperature (i.e. only 55 m from the entrance). This make the discussion in p. 8 lines 1-11 and 26-28 and p 9 lines 1-6and 8-24 rather speculative.*

- Cave temperatures were already determined in the past (e.g., by Kusch, 2000; see comment above). Thermal inertia of the host rock and monitoring results from comparable systems with a few 10 meters of overburden typically yield low temperature variability on the order of <±1°C inside the cave (Bunker Cave, Katerloch, etc.). Hermes Cave dips with about 28-30° downwards, which suggests low ventilation during summer time and some ventilation during winter.
- We'll amend the manuscript with information on temperature and ventilation

*Also, though I agree that clumped isotope results suggest a low degree of kinetic fractionation and that thus the speleothems δ18O can be a proxy for the δ18O of the precipitation, the calculation of drip water δ18O based on the modern temperature is not robust enough to be presented. I agree that rainfall amount can be the main driver of rainfall δ18O (and thus of speleothem) in this setting,as already shown by a number of studies and by precipitation monitoring in the region. However, this is not the only driver: changes in the seasonality would affect the final δ18O and are very difficult to quantify (as correctly stated), but also changes in the source of moisture can be important, though tricky to detect. As example, in mountain regions during summer there is usually a large proportion of moisture due to local evaporation and convective precipitation, whose isotopic composition cannot be simply related to amount effect. Also, during winter, the southern Balkans are interested by incursions of cold air from NW Asia, which correlated to increased snow cover, likely influencing the annual budget and thus the mean annual value of recharge. And there are many others variables...The influences of these effects are often difficult to disentangle and may be contrasting or may change during time (e.g. Dragusin et al., 2014).This means that a simple quantification of δ18O in terms of changes in the amount of rainfall is not straightforward and should be avoided in absence of a proper long-term monitoring of rainfall δ18O values at the specific cave locations. Please add more discussion about other potential effects and remove the quantification attempt (e.g. p4lines 4-5, pS1 lines 23-26, p 9 lines 15-16, p. 12 lines 23-32)*

- The manuscript will be rephrased as suggested. We reduce the aspects related to the quantification and add a detailed discussion about other potential effects on the oxygen isotope ratios.

- Based on IAEA isotope and rainfall data from Athens we investigated the influence of changing seasonality (Fig. S2a). We'll more prominently include this aspect in the discussion of the oxygen isotope data

**Interpretation of δ13C:**

*I found this part rather problematic. Honestly, I'm not able to see any common trend in curves presented in Fig. 5. δ13C andδ18O are not consistently anticorrelated as the paper claims, and to me their variations are largely disconnected. Also the proposed explanation for the supposed anticorrelation is not convincing at all, as δ13C values always remain in a range which is consistent with biogenic input from soil (see e.g.C3 Tremaine et al., 2011), moreover, the few examples of anti-correlation related to short infiltration times (Bar Matthews et al., 2003 but also Regattieri et al., 2018) consist in sharp opposite peaks in specific and restricted intervals of the records, and not to a slightly contrasting pattern in some points. I think the main drivers of δ13C are changes in the biogenic CO2 input and thus can be related to vegetation and soil state. The fact that they are not consistent with δ18O likely means that hydrological changes were not strong enough to deeply affect the soil state.*

- Supplementary Fig. S8 shows the (anti)correlation between δ$^{13}$C and δ$^{18}$O for the Hermes record as a whole. In a revised version we'll add figures focussing on specific time periods and rephrase the discussion along the lines as suggested by the reviewer

**Trace element interpretation:**

*Also this part is quite problematic. Again, I do not see any clear common trend between δ13C and P/Ca or between δ18O and Mg, and the interpretation of the P/Ca and Mg/Ca record is a bit odd and simplistic. P in speleothem has several potential origins, depending on individual cave settings. For example, where P-rich minerals like apatite are present, it can be sourced from the bedrock and be incorporated according to a distribution co-efficient in the crystal lattice as P-rich phases (Frisia et al., 2012). However, apatite is very rare. Instead, several works from temperate ecosystems indicate that phosphorus in cave drip water derives principally from the leaching of decomposing plant residue (Borsato et al., 2007; Treble et al., 2003). Thus, P concentration is often interpreted as proxy for infiltration and/or as a surface bioproductivity marker (Fairchild and Treble,2009), with variations related to changes in vegetation and soil conditions. The fact that here its variation are decoupled from those of the δ13C likely suggest the absence of major environmental variations throughout the studied period. Also the interpretation of Mg is not convincing. Due to its high solubility, it is mostly transported as solutes (Fairchild and Treble, 2009) and not within mineral detrital particle. A certain flux as solid is possible, but in this case I would aspect a stronger correlation with Al and Mn, whose variations instead are largely disconnected from that of Mg/Ca. The only thing that I see from the Mg record is that there is a slight similarity in its long term trend with that of theδ18O (both increasing), which may be due to hydrological variations (increased residence time during drier condition).*

- We complement the revised version with figures showing directly the relation between δ$^{13}$C and P/Ca as well as δ$^{18}$O and Mg/Ca and amend the discussion of the elemental ratios in a more comprehensive way
- We rephrase the discussion of the Mg/Ca ratios and focus on the long-term trends in consideration of potential hydrological variations

**U isotopes**

*Also here there are some problems in my opinion. For what I know, variations in speleothem [234U/238U]i can be related to changes in the relative proportion of U deriving from the carbonate bedrock versus that originating in soil (Kaufman et al., 1998;*

*Ayalon et al., 1999; Frumkin andStein, 2003; Hercman et al., 2020). Water interacting with more developed soil should have a higher 234U content, due to greater surface of mineral-water interaction. Thus, would aspect to have higher 234U/238U when the soil is more developed, i.e. under wetter conditions. Again, to me it points to a lack of major variations in soil and vegetation status during the studied period.*

- In the $^{234}$U/$^{238}$U discussion we include the mineral-water interaction of the soil and the bedrock and add related references. More developed soil under wetter conditions and higher $^{234}$U/$^{238}$U are consistent with our interpretation of wet and dry period in the Hermes Cave records and the measured U isotope values.

***4.2 event:*** *I agree that the oxygen record shows a hydrological change at that time (though not very prominent), but to me the other proxies do not, implying a very subdued expression at the cave site and a not very strong environmental response.*
*Comparison with other records: As correctly stated, the comparison between δ13C record from the Peloponnese does not shows any common trend, so in my opinion figure 8 could be avoided. I also found the correlation with others δ18O speleothem records a bit forced. I acknowledge some similarity in the general trend of some of the curves in fig. 9, but it is really really weak. Authors should be more honest in recognizing that most of the variations is not totally replicated, especially at the multicentennial scale. Also, why compare with the δ18O record from Sofular cave? This record mostly records changes in the proportion of moisture from the Med or the Black see, and it is not related to variations in the amount of rainfall!! The whole discussion from p 13 line 22 to p 14 line 22 about correlations with lake records etc make no sense because it is not supported by a figure. The readermust evaluate by itself the correlation among the records. The whole paragraph is advertising, not science....*

- The δ$^{18}$O variation at 4.2 ka is the strongest signal evolution of the whole record (>1‰) and occurs within a relatively short time period of ca. 100-150 years. The d13C variations are linked to it and support a significant climatic effect, although it seems to have had limited influence on the elemental ratios.
- δ$^{18}$O variations within the Holocene and in particular during the last 6 ka are generally small in speleothems and on the order of 1-2 ‰ (e.g., Bar-Matthews et al., 2003; Zanchetta et al., 2007; Fohlmeister et al., 2012; Demeny et al., 2019). Also stalagmite records from the Mediterranean region and the Middle East reveal rather limited variations in this time period (e.g., Cheng et al., 2019). In the context of these terrestrial records a variation of more than 1‰ appears comparatively strong
- The comparison of δ$^{13}$C of the selected speleothems will be reconsidered and potentially shifted to the supplementary
- We'll change the presentation of the records in Fig.9 and provide a direct overlay of the Hermes Cave records with the other records. The Sofular record is in particular interesting as it shows changes in the proportion of moisture from the Mediterranean or the Black Sea and can therefore give hints on the influence of air flow from NW Asia, constraining the interpretation of the δ$^{18}$O signal in Hermes Cave. We'll amend the manuscript by a corresponding paragraph
- Instead of current figure 8 we'll add a figure on the comparison with the discussed lake and terrestrial records

**Other points:**

*p1 line 22: change elemental ratios to trace element composition of to elemental/Ca ratios*
*p1 line 25: high degree of correlation is a bit too strong, I would say similarity*
*p2 line 11: Add However before A paleoclimate*

*p2 line 13: This sentence reads a bit odd, please rephrase like "Here we focus on a speleothem from Hermes Cave (NE Peloponnese, Greece) and compare our record with others climatic archivers, notably speleothems and lake sediments, from the same region" or similar*

*p2 line 23:Add a proper ref after conditions. I would also add that temperature quantification in stalagmite is often complex*

*p2 line 28-30 I would delete the sentence about the climate divide in the Peloponnese, as this point is not further discussed.*

- we'll rephrase the manuscript as suggested

*p 3 line 13: Is the vegetation totally composed of C3 plants?*

- We didn't perform a detailed botanical survey, but due to the high mountainous location C3 plants are likely dominating.

*p3 line 25-31it is not clear here or in the supplementary how the sensitivity of infiltration to temperature is calculated*

- The temperature effect on the infiltration was estimated with a first-order approach regarding the evaporation. We used the simplified connection between mean monthly temperature (Tm) and evaporation (E): E=2*T (E in mm). We'll add the corresponding information to the main manuscript and the supplementary and provide an assessment of the uncertainty with respect to more sophisticated approaches

*p3 line 32: Regarding infiltration as snow, I'm not very familiar with the specific cave setting, but I guess that most of the recharge is rain and not snow and that winter snow only lasts shortly, not enough to stratify in layers equilibrated and not equilibrated with the atmosphere.*

- We are not aware of a detailed long-term record on snow amount and duration of snow cover, however, the existence of a skiing center close to the cave site suggests sufficient precipitation as snow during winter time

*p4 line 7-12: Petrographical information must be added here. Also, the soot layer is mentioned here and in the abstract, but it is not further discussed in the following sections.*

- A thin section has been prepared and petrographically assessed. Related information will be added together with details on the soot layer.

*p 6 line 25-32 I appreciate this approach and this discussion, however there is a repetition with lines 25-31 of page 4, please collate the information here or in the methods only.*

- We'll remove the repetition

*p7 lines 11 and 15: add the proper±symbol.*

- will be changed as suggested

*p7 Elemental ratios: Are they molar ratios or simply element/Ca ratio with the Ca values considered invariant? Is not clear. Please explain and be consistent (and I would prefer just the element expressed in ppm and discussed as concentration, as the ratios are mostly used for elements in solution like Mg, Sr, Ba to discuss the occurrence of PCP. It would be interesting to see the value expressed as ppm, as in my experience very often elements such as Pb and Mn are very low in concentration and their changes not very significant.*

- We assumed that the Ca concentration is constant and assigned measured changes to analytical aspects. The elemental ratios are given in ppm/ppm. We'll add information on the concentration of each discussed element and how the measured values relate to the limit of detection.

*p7 line 26-27: I see just a slight similarity in the Al, Mn and Fe curve, not a clear correlation. A correlation table reporting r values between each pair of elements have to*

*be inserted, and also, it would be better to plot the ratios discussed together one close to each other (see comments on figures).*

- We'll add a correlation table

*p7 line 28-29: Given that I do not see any correlation between P,δ13C and δ18O, this information about the same number of peaks is not very useful (and also a bit misleading).*

- Will be reconsidered

*p7 29-30 The correlation between elevated Ba and Sr and Pb and Mn is very hard to evaluate from Fig. 5*

- The correlation between Ba/Ca and Sr/Ca is given in Supplementary Fig. S6. We'll add there also the relationship with Pb and Mn.

*p7 32: Can you quantify the shift and the associated chronological mismatch between the isotope and the. records?*

- There is no chronological mismatch between the different proxies as we correlated the proxy data following the growth layer. We amend Fig. 4 with the different analyses tracks (see Fig. R1)

*p8 line 2: Add the standard deviation between samples*

- the standard deviation of a single analysis was 0.017 ‰ for $\Delta_{47}$ based on the replication of reference material

*p 11 line 4-5: if Mg variations are related to hydrological variations, it makes no sense to calculate a related temperature change. Please remove. In general they are not of very good quality. Please enlarge the y axis and make the blue bands lighter, as they actually cover the curves and make difficult to evaluate them.*

- We'll update the figures and remove the temperature calculation in relation to the Mg/Ca ratios.

**Additional references:**

Boch, R., Spötl, C., Frisia, S., 2010. Origin and palaeoenvironmental significance of lamination in stalagmites from Katerloch Cave, Austria. Sedimentology, doi: 10.1111/j.1365-3091.2010.01173.x

Riechelmann D. F. C., Schröder-Ritzrau A., Scholz D., Fohlmeister J., Spötl C., Richter D. K. and Mangini A. (2011) Monitoring Bunker Cave (NW Germany): a prerequisite to interpret geochemical proxy data of speleothems from this site. J. Hydrol. 409, 682–695.

Kusch, H., 2000. Die "Hermes-Höhle" auf dem Ziria-Massiv, ein korinthisch-hellenistischer Kultplatz (Peloponnes, Griechenland). Die Höhle, 2, 52-63.

**Additional Figure**

[Figure]

**Figure R1**: Spatial relation of various proxy analyses. The clumped isotope samples (deeper pits) were directly taken on the stable isotope ($\delta^{13}$C, $\delta^{18}$O) sample track that followed the visible stalagmite layering. The elemental ratios were analysed parallel to the isotope track. Samples for U/Th analyses were taken as thin blocks at the right part of the stalagmite slab. All proxies were correlated to the depth axis using the stalagmite layers as orientation.

---

## Author Comment (AC2) · 21 Sep 2020

**Reply to anonymous referee #2**

In the following we reply to the main point of criticism. Minor aspects (e.g., related to language and grammar) will be directly considered in a revised version. Referee comments are in italic and grey, the author response in black

*Kluge et al present a new multiproxy speleothem record from Peloponnese ranging from ca. 0.8 to 4.7 ka with the aims to give new data on climate evolution to place societal and cultural evolution of this important part of the Mediterranean during Bronze and Iron age. Despite this ambitious target, the manuscript basically fails to obtain important insight on it, because the high clastic contamination produce a large spread of U/Th ages and associated error, and the signal of many proxies is not always pronounced.*

- The comparison to other speleothems chronologies from the Peloponnese and Greece shows that the Hermes Cave stalagmite has age uncertainties comparable to other records from the region. For example, the stalagmite chronology of Finné et al. (2014) from Kapsia Cave (Peloponnese) has age uncertainties of ca. 170 to 1600 a.  Depending on the detrital contribution lower age uncertainties on the order of a few decades may be achieved for the Holocene (e.g., Finne et al., 2017). This is also visible for GH17-04, for which the 4.2 ka event was dated with reasonable uncertainties for low-uranium speleothems of 70-130 a.
- Additional U/Th analyses will further constrain the chronology. We'll address in particular the oldest and the youngest part of the stalagmite
- The proxy signals reflect the high alpine environment and are therefore potentially attenuated regarding  rainfall amount variations and, based on the clumped isotope results, not enhanced by kinetic effects

*The authors are aware of this and at certain points of the manuscript tried to focus on a short interval close or corresponding to the so called 4.2 event. Also in this case the discussion is not able to focus on substantial new ideas on this period. So the general aspect of the manuscript is confused in the treatment of the data in the discussion and then in the abstract and the conclusion.*

- The most pronounced and clear signal of the whole d18O record occurs around the 4.2 ka event. In the light of the ongoing discussion on the nature of the 4.2 ka event and the previous special issue on this topic in *Climate of the Past* we deem it important to add and discuss related data that is unique for Greece
- A more coherent discussion with adjusted abstract and conclusion will be prepared

*First of all the manuscript needs a complete reorganization in the aims and introduction. The present aims are good for a speleothems with better chronology and much better signal. Along the manuscript, we pass in a confounding way (some time with repetitions) between the description of the long-term record and the short term changes. These are some of the generalcomment I have. However, there are many points along the manuscript, which needs*
*to be improved. I try to give some example below.*

- We'll improve the chronology by additional radiometric analyses and check if a statistical robust correlation is possible with other, better dated stalagmite records from the Peloponnese or Greece in general
- The discussion will be oriented at the revisited uncertainty of the chronology and correspondingly restructured and rephrased

*Abstract Pag. 1 Line 15 the 3.2 even is mentioned, and in many figures is highlighted but along the text I never see a serious discussion on that. Moreover, the chronology of this interval is really poor.*

- We remove the reference to this event and the highlighting in the figure

*Pag. 1 Line 20 the record is reported continuous between 800 and 5300 ka differently from the conclusion*

- Typo to be corrected

*Pag. 1 line 24 234U/238U: there is no particular discussion on this point along the text to be so relevant to be mentioned in the abstract (indeed is lacking in the conclusion).*
- Will be removed from the abstract

*Introduction Pa.g 2 lines 14-15. I think we must be aware that speleothems can be precisely dated if clastic contamination is negligible.*
- We add the restriction that the absence of significant detrital contamination is an important condition for the possible high-precision radiometric dating of speleothems

*Pag. 2 lines 30-31 This the style of the manuscript a description of a short event and then a focus on long term trend. So, most of the introduction is not useful to justify this view.*
- We restructure and rephrase introduction and discussion accordingly to match the description of events on various durations

*Study Area*
*Pag. 4 lines 2-5 The data from Nehme et al., 2019 and Bar-Matthews et al.,*
*2003 cannot be used acritically for Peloponnese. They cannot be presented as valid*
*data for your area. There is a general "paradigmatic" view on the interpretation of the*
*d18O in the Mediterranean and I can agree this can be used for past reconstruction.*
*Moreover, in your discussion you try to justify this view using also other proxies. I think*
*this is a correct "qualitative" approach. In absence of regional-to-local convincing data*
*on precipitation to show data from other sector is not good.*

- We embedded our presentation and discussion of the oxygen isotope ratios in rainfall in the larger regional context and also cite the IAEA-WMO data base. In the supplementary we show recent isotope data of Athens (IAEA-WMO) that also indicates a rainfall amount-d18O relationship for Athens. The relatively close proximity of the Peloponnese to Athens (about 100 km to Hermes Cave) justifies its use for interpretation at the cave side
- The negative d18O-rainfall relationship holds for most of the Eastern Mediterranean (s. Nehme et al., 2019; Bar-Matthews et al., 2003; IAEA-WMO), but should be investigated also for the cave site. As we are missing local hydrological and long-term isotopic data for the Hermes Cave site we'll amend the text by a critical assessment

*Material and Method*
*Pag. 4 line 10 "a soot layer: : :...can you please show the position in the figure 4. In the*
*text there is no mention of thin section and just a brief description on the fabric would*
*be useful also for the equilibrium conditions and to discuss if there are hiatuses. In*
*some points it seems likely.*
- We'll highlight the soot layer (already visible as blackish layer at about 15mm from top) and add a photograph of the thin section, including discussion of fabrics and a discussion on possible hiati

*Pag. 4 Line 17 ": : :where manually pre-treated to obtain pure carbonate: : :.." please can you explain more precisely?*
- To be deleted (the speleothem carbonate is already pure)

*Pag. 4 line 25. Clastic contamination can be also related to clastic-carbonate? I'm not*
*an expert on U/Th measurements.*
- Contamination of clastic carbonates is in general possible but unlikely for speleothems. Detrital contaminants are flushed in with the drip water and typically relates to clay minerals.

*Results Pag. 6 line 24 "may be"? "is" better.*
- Will be changed as suggested

*Pag. 7 lines 5-10. It Is unclear why you use two different Bayesian program and then choose one instead the other. Can you show both?*
- Already shown in supplementary figure S5
- Depending on the algorithms implemented in the program, slight differences in the chronologies may occur

*Pag. 7 line 14. "suggest" I understand what the authors want to say, considering the large error, but I prefer "indicate".*
- Will be changed as suggested

*Pag. 7 line 19. Can you show this trend with a polynomial curve? Can you be statistically confident that this a trend or is just a visual impression? Please can you explain why in figure a different averaging is choose for Skala Marion. Can you show with a polynomial curve the trend described in the text of the Hermes Cave?*
- We'll add a supplementary figure focussing on the longterm trends with a statistical discussion and polynomial curves

*Pag. 8 Lines 5-8 ": : :the general correspondence of individual and average: : :.." Considering there is only one single measure of cave temperature and the large T variability obtained using clumped isotopes and the associated error the conclusion would be: there is no secure conclusion.*
- Clumped isotopes are in general very sensitive to disequilibrium (references in the manuscript). Temperature offsets in case of disequilibrium could be on the order of 10°C and up to 40°C and are therefore clearly detectable.
- Measured clumped $\Delta_{47}$ values scatter and have partially elevated uncertainties. Calculated temperature are, however, with one exception all within the range of modern cave temperatures (note that there has been another instrumental cave temperature measurement in 1996, see response to ref.1). The average temperature of all clumped analyses over the speleothem growth period is 8.5±1.4°C (n=8, 1SE), well corresponding to the modern temperature measurements. Considering the small uncertainty of the average value of the full data set, the correspondence between D47-based temperature and cave T suggest no or negligible influence of disequilibrium

*Discussion Pag. 8 lines 14-15. If the manuscript is focused on this interval this should declared since the introduction and the manuscript structure should be mostly different and most focused. But the general organization of the manuscript is not well done. There is no a clear focus. At the end what do the authors want solve? What do they have then solved?*
- Speleothem proxy signals provide an insight into paleoclimate and environmental changes. Depending on the chronology and the time resolution the short term signals can be discussed (around 4.2 ka, reasonably well constrained chronology) or the long-term evolution investigated (rest of the record)
- We provide continuous information on paleoclimatic changes for about 4 ka during the most interesting period of cultural evolution in Greece, complement existing speleothems records both temporally and spatially and add unique and detailed information on the 4.2 ka event in this region
- We'll reorganize the structure and clearly outline the focus of our discussion

*Pag. 8 lines 15-17. These two sentences are rather confounding. The chronological uncertainties are elevate or most of the record and not just on top. A detailed correlation with historical event is honestly not applicable (if we can exclude a brief interval). Indeed the second sentence is correct.*
- will be rephrased

*Pag. 8 lines 27-28. Once again I don't think to stress to this point is useful.*
- it is a very important aspect for the interpretation and discussion of the δ¹³C and δ¹⁸O data. Furthermore, it impacts also the discussion of elemental ratios as PCP can have a major influence (e.g., on Mg/Ca ratios). Indications for the absence of PCP constrains the remaining causes and allows a more robust discussion.

*Pag. 8 lines 28-31. I don't think that the conclusion of Borsato et al. (2016) can be transported acritically from Alps to Peloponnese in a so strict sense without a general monitoring program like the data presented by Borsato et al.*

- we'll add a note of caution and rephrase the corresponding sentences

*Pag. 9 lines 4-6. The ranges of values is quite large. In absence of more detailed local data many calculation are probably misleading.*
- We are intending to amend our data set with drip water samples from the Hermes cave site.
*Pag. 9 lines 15-16 The relationship reported by Bar-Matthews is very local, and it cannot used for Peloponnese.*
- see response regarding "study area" above

*Pag. 9 lines 20-24 this is one of the few point where other short term oscillation are considered.*
- the discussion will be restructured to better represent the main topics of the manuscript and to focus the content

*Section 5.2 There are a lot of literature on 4.2 event, but the discussion proposed did not add relevant points.*
- Absolute dated and reasonably well-resolved signals of the 4.2 ka event are rare. For Greece and the Peloponnese a corresponding signal has not yet been reported and is therefore a novelty.

*Line 28 Rousseau et al., 2019 is not a paper. So difficult to quote.*
- removed

*Pag. 12 lines 23-27 here there are some sentences and concept repetitions. Once again, I think is quite misleading to use the correlation defined for far areas.*
- based on the IAEA data the rainfall amount-d18O relationship has been observed throughout the Eastern Mediterranean and is also visible for Athens. See supplementary figure S1. We'll rephrase the text to clarify the relationship and to indicate that it does not only hold for the Levant.

*Pag. 14, lines 14-16. It is hard to say that Mavri Trypa provides a similar climate picture.*
- We'll be more specific and clearly indicate in detail similarities and discrepancies.

*Pag. 14, lines 20-21. "Furthermore, both records show a high degree of consistency in medium and hig-frequency fluctuation." Absolutely not. This is an overexploitation of the data. Moreover, there is a mention to high-frequency oscillations which have not been discussed in detail along the manuscript.*
- The data of Alepotrypa Cave are so far only available as a thesis (Boyd, 2015). We'll therefore remove the comparison. Note, however, that a comparison of both records shows indeed a high degree of overlap for the longterm trends and even higher frequency fluctuations (allowing for wiggle matching within the dating uncertainty).

*Pag. 14 lines 22-30. In some part of the manuscript the correlation with Lake Stymphalia is presented as strategic for the general interpretation. There are no proxy records show for this lake and the "comparision of the trends: : :.is difficul". There are other lakes cited but the record are not shown. This section seems quite useless.*

- The revised version will be complemented by a figure with direct comparison of the Hermes Cave proxy record with the other discussed lake records

*Pag. 14, lines 1-13 There is a discussion of records which are not show in any figure, so the comparison is difficult.*

- We'll complement the figures with all records that are discussed in the text

*5.3 implications*

*Pag. 14 lines 23-24. This point appears here for the first time and there is no any discussion. This section is not "implication" but already a summary of the main result, some not discussed at all, like the list of drier events reported as last point at pag. 15.*

- we'll restructure this section, add a discussion of all points that were not addressed before and move the list of drier periods to the conclusion

*Conclusion*
*Pag. 14 line 14. Where along the manuscript do emerge that there is a cooling trend? The introduction promise some conclusion related to social evolution and climate, but then?*

- Reference to cooling trend will be removed
- we'll add an extra paragraph on the inter-relation between social evolution and climate and what our study could contribute to this topic

*Overall, I consider the manuscript not suitable for publication even if the data can have some interest. I suggest to change the target of the manuscript, basically deciding which is the main focus and what wants to solve really and not what would be interesting to solve. The chronology is relatively poor so, an honest and calibrated manuscript is necessary and in this case, for me, welcome*

- we are sorry to hear that the manuscript in its current form is not ready for publication but are grateful for the detailed suggestions on possible improvements

---

## Author Comment (AC3) · 21 Sep 2020

**Reply to anonymous referee #3**

In the following we reply to the main point of criticism. Minor aspects (e.g., related to language and grammar) will be directly considered in a revised version. Referee comments are in italic and grey, the author response in black

*The manuscript presents a 4000-long palaeoclimate record from Late Holocene, from Hermes Cave (Peloponnese, Greece). The area and the period is obviously very interesting for archaeological, geoarchaeological and societal sciences too. During the last years/last decade, there have been very interesting, robust and solid studies of this period and in the general area of East Med, Aegean and Greece in particular. The level of current knowledge is quite high and any new addition should step on this and make a step forward. The major problem of the study by Kluge et al. is that their proxies, together with their assessment of the data, are not supporting their conclusions. It seems that the authors tried to use their available data and forces some conclusions which cannot be supported. The manuscript is well written and the methods used are indeed enough to proceed to a palaeoclimate study, but the results are not helping. It is probable that the specific speleothem archive is actually not suitable for such a study.*

- So far, there exists no continuous speleothem record for the last 5000 years for the Peloponnese. Furthermore, the existence of several records from the same region would allow to assess signal reproducibility and climatic relevance and, given the temporal resolution is comparable, also to investigate potential spatial pattern. In this respect, the Hermes Cave stalagmite can complement the fragmented stalagmite data from the Peloponnese. In a revised version we'll strengthen the regional comparison and embed the new results in a more comprehensive discussion

*Major remarks: - Age uncertainty. The stalagmite growth period is very short, so the age model must be very precise and accurate. Unfortunately, the GH17-05 calcite was not 'clean' enough to give proper U/Th datings for this purpose. It is impossible to proceed to such detailed and specific interpretation, discussion and reconstruction with such uncertainties in Late Holocene.*

- See response to ref. 2
- The stalagmite growth period covers almost half of the Holocene which is related to the major phase of societal development in Greece
- The age model will be improved by additional radiometric dates and comparison with well-dated speleothem sequences from Greece
- The uncertainties are variable, but comparable to other speleothem records with low uranium content. Especially around the 4.2 ka event the chronology is reasonably precise and accurate and allows to provide unique, new information

*- Identification of phases. The authors proceed to identifying phases of wet and dry pulses, based on vague observations on the d18O curve. These observations must be justified, by any means of analysis, considering value trends, statistical evaluation (is a pulse an outlier from the rest of the curve?), normalization or any other way of analysis. In some proxies, a periodicity is given (eg. line 28, page 9) without any analytical calculation (naked eye?).*

- A sub-section about statistical methods will be added in the revised version

*- Discussion. Correlation with other records is not properly justified. The observations by the authors are not really visible in the plots, even by comparing some of the records and excluding others. One needs to keep in mind the age uncertainties as well, in order to try to find correlating pulses between records. Eg. detecting dry conditions in time windows of 0.1 ka (eg. line 8, page 15) is not consistent with the age model.*

- We'll restructure the discussion regarding the comparison with the other records and also visually support our observations in the corresponding figures

*- Figures. Figures need reorganization and improvement. Fig. 1 A, does not give a clear location of the cave. Fig. 1 B, is not really needed in such details. Fig. 2 B, photo is not helping actually. Fig. 4, the stalagmite needs more info plotted, such as the axis of sampled positions. Fig. 8 and 9, the references of the presented records are missing, the should be added here and in the reference list.*

- Suggested improvements will be implemented in the revised version

*Generally, the article cannot be accepted in its present form. It needs an overall major revision and additional analytical effort (eg. age model) in order to be considered for publication in CoP.*